# Connecting genomic results for psychiatric disorders to human brain cell types and regions reveals convergence with functional connectivity

Shuyang Yao [1,2], Arvid Harder[1,2,9], Fahimeh Darki[3,9], Yu-Wei Chang[4], Ang Li [5], Kasra Nikouei[1], Giovanni Volpe [4], Johan N. Lundström [3,6], Jian Zeng[5], Naomi R. Wray [5,7], Yi Lu [2], Patrick F. Sullivan [2,8] ✉ & Jens Hjerling-Leffler [1] ✉

Identifying cell types and brain regions critical for psychiatric disorders and brain traits is essential for targeted neurobiological research. By integrating genomic insights from genome-wide association studies with a comprehensive single-cell transcriptomic atlas of the adult human brain, we prioritized specific neuronal clusters significantly enriched for the SNP-heritabilities for schizophrenia, bipolar disorder, and major depressive disorder along with intelligence, education, and neuroticism. Extrapolation of cell-type results to brain regions reveals the whole-brain impact of schizophrenia genetic risk, with subregions in the hippocampus and amygdala exhibiting the most significant enrichment of SNP-heritability. Using functional MRI connectivity, we further confirmed the significance of the central and lateral amygdala, hippocampal body, and prefrontal cortex in distinguishing schizophrenia cases from controls. Our findings underscore the value of single-cell transcriptomics in understanding the polygenicity of psychiatric disorders and suggest a promising alignment of genomic, transcriptomic, and brain imaging modalities for identifying common biological targets.

Genome-wide association studies (GWAS) have yielded fundamental insights into the nature of a wide range of human diseases, disorders, biomarkers, and traits. A recent summary[1] of 4593 GWAS publications studying 3908 phenotypes found 156,556 significant SNP-trait associations; notably, only 4.19% of significant SNPs were in a protein-coding region. GWAS have been particularly informative for psychiatric disorders whose enigmatic nature has long impeded progress. This body of work has shown that major psychiatric disorders are heritable, that clinically dissimilar disorders nonetheless have genetic overlap, and that this can clarify causality[2–6].

However, the genetic architectures of psychiatric disorders have proven to be particularly complex[7]. For example, predictions that genomic studies of schizophrenia would readily identify a few genes with near-causal effects[8–10] are inconsistent with the accumulated results: empirical studies of common genetic variation, rare copy number variation, rare exonic variation (both de novo and inherited),

[1]Department of Medical Biochemistry and Biophysics, Karolinska Institutet, Stockholm, Sweden. [2]Department of Medical Epidemiology and Biostatistics, Karolinska Institutet, Stockholm, Sweden. [3]Department of Clinical Neuroscience, Karolinska Institutet, Stockholm, Sweden. [4]Department of Physics, University of Gothenburg, Gothenburg, Sweden. [5]Institute for Molecular Bioscience, University of Queensland, Brisbane, Australia. [6]Monell Chemical Senses Center, Philadelphia, PA, USA. [7]Department of Psychiatry, University of Oxford, Oxford, UK. [8]Departments of Genetics and Psychiatry, University of North Carolina, Chapel Hill, NC, USA. [9]These authors contributed equally: Arvid Harder, Fahimeh Darki. ✉e-mail: pfsulliv@med.unc.edu; jens.hjerling-leffler@ki.se

and whole genome sequencing[11–14] were well-powered to detect a few causal genes shared by most cases and yet none were identified. Compared to many other human diseases/disorders, schizophrenia is notably polygenic[15] with the major population impact resulting from inheritance of a large number of common variants of small effect[12,14,16]. Indeed, the most recent GWAS for schizophrenia[14] implicated 287 genomic loci (median size 652 kb, interquartile range, IQR, 238–652 kb) often intersecting multiple protein-coding genes (median 2, IQR 1-6, and 13% of all loci contained no protein-coding genes). It is highly likely that there are many more loci to be discovered. It is not clear how these findings inform understanding of the fundamental nature of schizophrenia or what the underlying neurobiology might be.

In this paper, we evaluate the evidence that genomic results for brain disorders, diseases, and traits point at specific brain cell types. We evaluate the overarching hypothesis that cell types are an important readout of genomic studies for notably complex psychiatric disorders. We[17,18] and others[19–26] have previously evaluated this idea. However, for brain traits, the key limitations have been the sheer cellular complexity of the brain and limited transcriptomic data that previously forced reliance on mouse brain transcriptomic surveys. Siletti et al.[27] recently published the most comprehensive human transcriptomic dataset to date: single-nucleus RNA sequencing (snRNAseq) of 3.369 million nuclei from 106 anatomical dissections within 10 brain regions.

Here, we incorporate this large-scale human brain atlas along with newer and larger GWAS. We extend our prior work by evaluating evidence for anatomical regions as well as their functional connectivity. Cell types form local networks that are connected to different brain regions. Functional magnetic resonance imaging (fMRI) is a non-invasive and widely used tool to evaluate brain regional functional connectivity in both health and disease. Systematic reviews have suggested disturbances in the Default Mode Network and the Core Network in cases with schizophrenia and their neurotypical relatives[28–30]. These findings suggest that genetic liability to schizophrenia can be manifest in empirically-defined cell types, anatomical regions, and in functional connectivity between brain regions. Despite the challenges in analyzing the high-dimensional fMRI data[31], in this paper, we integrate fMRI data from schizophrenia cases and controls to illustrate how genetic information, via transcriptomics, agrees with fMRI in the prioritization of brain regions and changes in functional connectivity using two independent fMRI data sets. Identifying affected brain regions and circuits is important given "interventional psychiatry" therapeutics that can modulate activity of specific brain regions (e.g., transcranial magnetic stimulation or deep brain stimulation).

## Results

Our overarching goal was to evaluate whether the genomic regions identified by GWAS for complex brain phenotypes implicated specific brain cell types, anatomical regions, or their functional connectivity. As diagrammed in Fig. 1, we integrated the most comprehensive human snRNAseq brain atlas to date[27] with GWAS summary statistics for 36 primary traits including psychiatric disorders, brain traits, neurological diseases, structural MRI measures, and control traits (Supplementary Data 1, Supplementary Fig. 1, and *Methods* for inclusion criteria). We systematically processed summary statistics for these GWAS. Supplementary Fig. 2 shows the genetic correlations between the primary traits which were in accord with prior reports[3]. As in our past papers[17,18], we used stratified LD score regression (S-LDSC) to estimate the enrichment of SNP-heritability for a trait in genes whose expression typified cell classes. The genetic liability of a trait can be measured by SNP-heritability, the proportion of phenotypic variance in a trait attributable to the additive genetic variation estimated from GWAS data[32].

Cellular diversity is hierarchically organized in the brain[33,34], from a tripartite classification (neuronal excitatory, neuronal inhibitory, and non-neuronal) to higher-order cell superclusters that are divisible into clusters and subclusters/cell types. Following the Siletti nomenclature[27], we analyzed 31 superclusters and their component 461 clusters. In Supplementary Data 2, we characterize the superclusters: 10 non-neuronal and 21 neuronal superclusters (13 excitatory, seven inhibitory, and one mixed neuronal supercluster). The supercluster labels capture major features but, inevitably for a complex tissue, some labels do not capture all features: for instance, "medium spiny neurons" and "eccentric medium spiny neurons" also contain cells from outside caudate and putamen (e.g., other long-range projecting inhibitory cells) and "amygdala excitatory neurons" also contain cells from paleocortex. Non-neuronal superclusters generally derived from dissections across the brain: e.g., astrocyte, ependymal, fibroblast, oligodendrocyte, and vascular cells were identified in many anatomical regions. Neuronal superclusters usually had a main anatomical region: e.g., deep-layer near-projecting and upper-layer intra-telencephalic excitatory neurons were from neocortical dissections and the three hippocampal cell classes were from hippocampus (the chief exceptions were the more heterogeneous miscellaneous and splatter superclusters).

We identified protein-coding genes whose expression was highly specific for each brain cell type as assessed by top decile expression proportion (TDEP or "gene specificity"). Li et al.[35] determined that the TDEP approach combined with S-LDSC had power and false positive rates that were jointly equivalent or superior to eight other methods (*Methods*). We also compared gene selection using relative versus absolute expression (i.e., TDEP versus TPM, transcripts per million), and found that both yielded nearly identical high-dimensional visualizations (Supplementary Fig. 3) but TPM was strongly influenced by broadly expressed "housekeeping genes" and this altered (and in some instances biased) gene ontology (GO) gene set analysis results (*Methods*).

We posit that TDEP genes for cell types are enriched for biological processes related to cellular identity and function. First, TDEP genes for different cell types generally had low overlap (Supplementary Fig. 6, median Jaccard index 0.049, IQR 0.022–0.099) but certain pairs had greater overlap (necessitating conditional analyzes, Fig. 2A). Second, as expected, "housekeeping" genes (highly and consistently expressed across tissues)[36] were markedly less likely to be TDEP genes. Third, we conducted GO gene set analyzes[37] for lists of the -1300 TDEP genes per supercluster (Supplementary Data 2). Results for non-neuronal cells suggested a diverse range of significant GO terms consistent with the supercluster labels: astrocyte with biological adhesion, choroid plexus with cilium, microglia with immune response, oligodendrocyte with neuron ensheathment, oligodendrocyte precursor with gliogenesis, and vascular with vasculature development. GO terms for most neuronal cell types were dominated by synaptic biology, consistent with findings that forebrain neuronal cell identity is to a large extent driven by specific expression of synaptic genes[38] (exceptions were lower/upper rhombic lip and cerebellar inhibitory neurons from non-cortical regions). Fourth, the "*Methods*" section describes additional features of TDEP genes: (a) visualization of supercluster TDEP genes yielded groups of non-neuronal cells, neocortical excitatory neurons plus medium spiny neurons, and inhibitory interneurons plus non-cortical excitatory interneurons (Supplementary Fig. 4A); (b) TDEP genes tend to co-occur in genomic regions (Supplementary Fig. 5); (c) visualizations for gene expression specificity and genomic co-occurrence were similar suggesting that TDEP genes tend to be located near each other; and (d) all neuronal TDEP genes accounted for 61–65% of the SNP-heritability for the largest brain trait GWAS (scz2020, bip2021, mdd2019*, neuroticism, education, and IQ).

### Identifying human brain cell types implicated by GWAS

For each of the 36 primary GWAS, we estimated SNP-heritability enrichment for TDEP genes in each of the 31 superclusters (1116

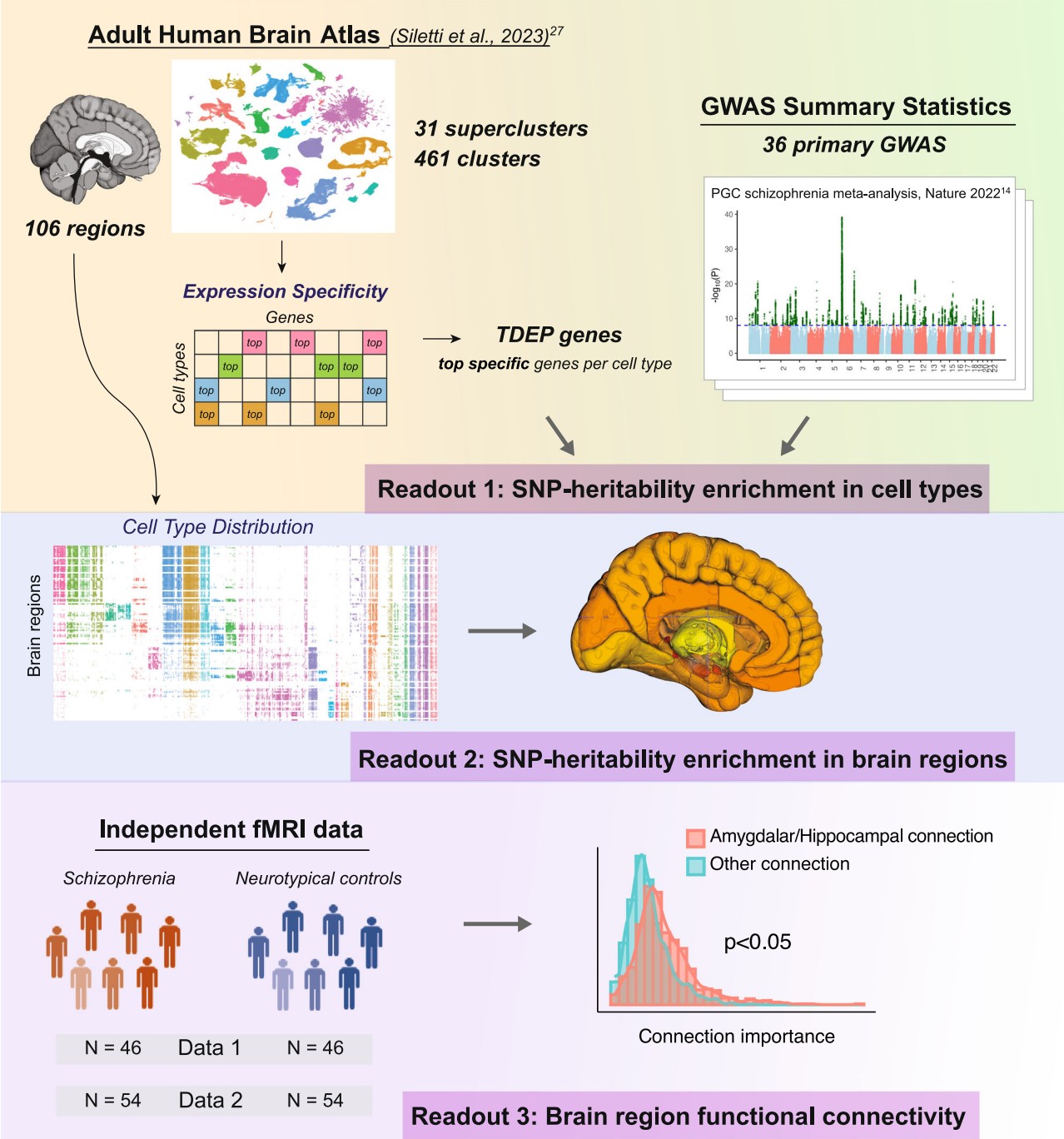

**Fig. 1 | Study schematic.** We first identified cell types enriched for the SNP-heritability of 36 primary traits including major psychiatric disorders, using the most comprehensive Adult Human Brain Atlas. This was integrated with the cell type distribution across brain regions to identify brain regions enriched for the SNP-heritability of the traits. Finally, the regions suggested by cell-type-informed SNP-heritability enrichment were used to explore brain region functional connectivity that can differentiate schizophrenia from neurotypical controls in two independent datasets. TDEP=top decile expression proportion, which were the most specifically expressed genes in each cell type (supercluster or cluster).

estimates). We used FDR correction for multiple comparisons per GWAS (Fig. 2A, Supplementary Data 3). The P-values were not uniformly distributed (modes near 0 and 1) and 9.6% of all comparisons had FDR < 0.05. Of the non-neuronal superclusters, only microglia reached significance for any trait (lymphocyte count, neutrophil count, and multiple sclerosis). As in our prior reports[17,18], non-neuronal superclusters were not significant for psychiatric disorders. In contrast, eight neuronal superclusters accounted for 60% of all significant enrichments. The numbers of superclusters with significant trait SNP-heritability enrichments were highly variable: (a) none of the structural

MRI measures; (b) most neurological diseases had none (except for multiple sclerosis and epilepsy); (c) of the control traits, neutrophil and lymphocyte counts enriched for microglia and BMI enriched for two deep layer pyramidal cell superclusters and amygdalar excitatory neurons; (d) broad neocortical and non-cortical signals are also observed for other brain traits including educational attainment, IQ, neuroticism, and alcohol drinks per week with all neuronal neocortical superclusters significant for scz2022 and bip2021 (except for deep-layer near projecting). Non-cortical forebrain clusters from "amygdala excitatory" to "eccentric medium spiny neuron" showed strong signals

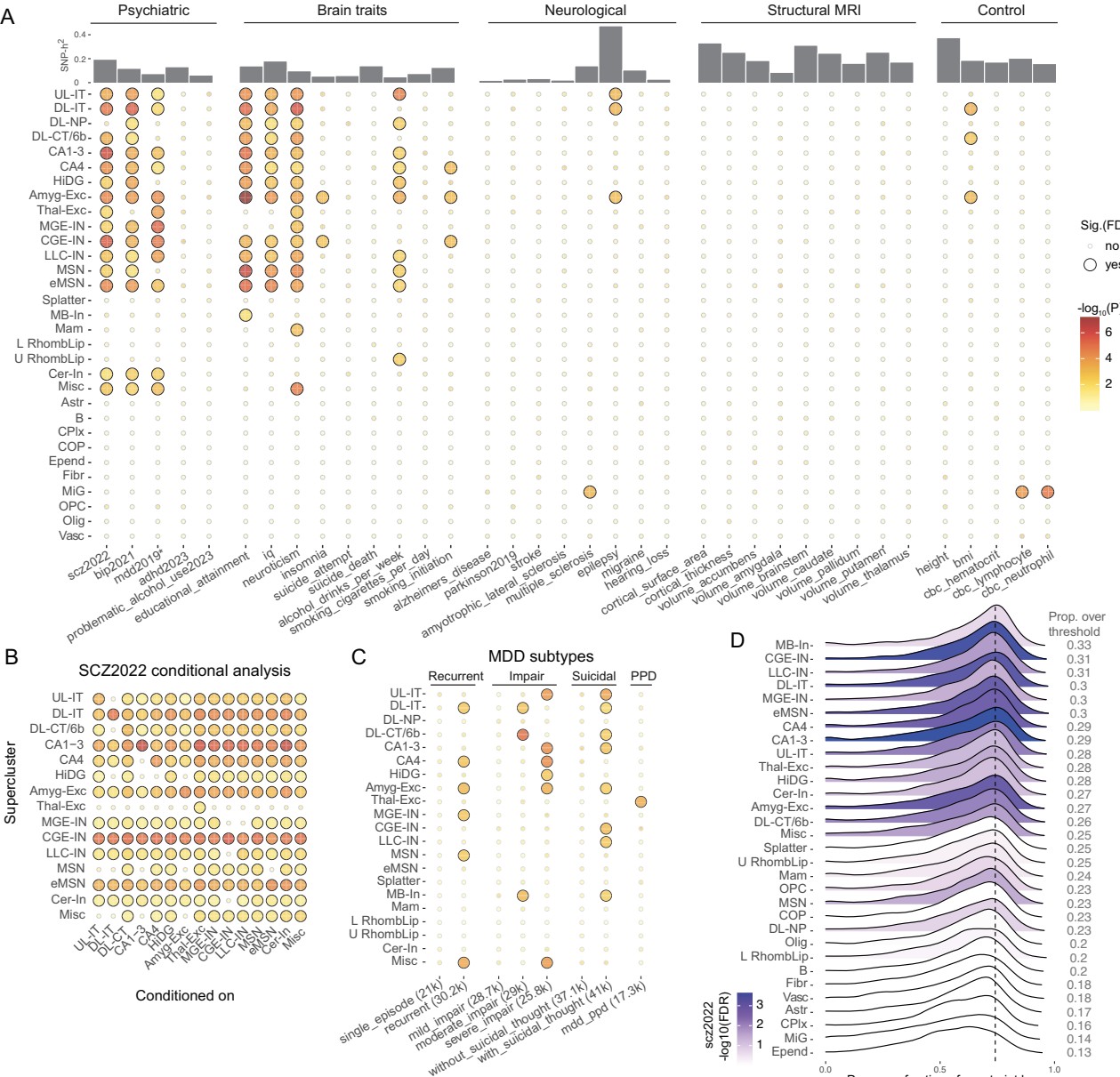

**Fig. 2 | Supercluster results. A** SNP-heritability enrichment in 31 superclusters for five phenotype categories. One-sided *P* value was calculated based on the LDSC coefficient z-score, and dot color indicates enrichment significance as –log10(P) with darker reds indicating greater significance. Large dots indicate enrichment significance adjusted for multiple comparison at FDR ≤ 0.05. We interpret the suicide results cautiously as this GWAS did not clearly control for MDD[93] which may confound the enrichment pattern. mdd2019* summary statistics did not include data from 23andMe. Abbreviations of superclusters: UL-IT: Upper-layer intratelencephalic. DL-IT: Deep-layer intratelencephalic. DL-NP: Deep-layer near-projecting. DL-CT/6b: Deep-layer corticothalamic and 6b. CA1-3: Hippocampal CA1-3. CA4: Hippocampal CA4. HiDG: Hippocampal dentate gyrus. AmygExc: Amygdala excitatory. ThalExc: Thalamic excitatory. MGE-IN: MGE interneuron. CGE-IN: CGE interneuron. LLC-IN: LAMP5-LHX6 and Chandelier. MSN: Medium spiny neuron. eMSN: Eccentric medium spiny neuron. MB-In: Midbrain-derived inhibitory. Mam: Mammillary body. L RombLip: Lower rhombic lip. U RombLip: Upper rhombic lip. Cer-In: Cerebellar inhibitory. Misc.: Miscellaneous. Abbreviation of non-neuronal superclusters: Astr: Astrocyte. B: Bergmann glia. CPlx: Choroid plexus. COP:

Committed oligodendrocyte precursor. Epend: Ependymal. Fibr: Fibroblast. MiG: Microglia. OPC: Oligodendrocyte precursor. Olig: Oligodendrocyte. Vasc: Vascular. **B** Conditional analysis of superclusters for enrichment of scz2022 SNP-heritability. The y-axis is the supercluster of interest and the x-axis indicates the supercluster conditioned upon. For convenience, the unconditional results are on the diagonal. Full results are shown in Supplementary Data 4. **C** SNP-heritability enrichment for MDD subtypes. Number of cases is shown in parenthesis in the x-axis label. Non-neuronal superclusters did not have signals for any subtype and were therefore omitted in the plot. Dot color and size are the same across panels A-C. (**D**) Ridge plot showing the density of the evolutionary constraint for TDEP genes of each supercluster. For each gene, the proportion of constraint 1 in its CDS bases was used as the measure of evolutionary constraint. The vertical dashed line shows the 80th percentile for evolutionary constraint for all protein-coding genes. The right column gives the proportion of TDEP genes above the 80th percentile of constraint. The plots were colored by the SNP-heritability enrichment for scz2022 (-log10FDR).

for both scz2022 and bip202 (Fig. 2A); and (e) the significant enrichments were dominated by six complex psychiatric disorder/brain traits. For the largest and most powerful GWAS traits (scz2022, bip2021, mdd2019, neuroticism, education, and IQ), the same eight

neuronal superclusters had significant enrichment for all six traits. This agrees with the observation that schizophrenia, bipolar disorder, and MDD account for substantial morbidity and mortality and have considerable clinical and pharmacotherapeutic overlap (especially for

severe and enduring forms of illness). Moreover, IQ, educational attainment, and the "Big 5" personality trait of neuroticism are important patient stratifiers and/or modifiers of clinical course[39–41]. The neuronal superclusters were five excitatory (amygdala excitatory, deep-layer intratelencephalic, hippocampal CA1-3, hippocampal CA4, and upper-layer intratelencephalic) and three inhibitory (CGE interneuron, eccentric medium spiny neuron, and LAMP5-LHX6 and chandelier).

**Excluding alternative explanations.** We evaluated a set of potential explanations for the observed overlap of eight superclusters with six GWAS traits. First, the genome-wide genetic correlations between the six traits were occasionally high but far from complete: for the 15 unique genetic correlations, the median $|r_g|$ was 0.22, IQR 0.15–0.39. The largest $r_g$ values were 0.73 for educational attainment-IQ, 0.69 for mdd2019-neuroticism, and 0.68 for bip2021-scz2022, and all other values were < 0.5. Second, the significant GWAS loci for these six traits only infrequently intersected with GWAS loci of more than one trait (median Jaccard index 0.069, IQR 0.034–0.094). Third, TDEP genes did not have fully explanatory overlap: of the 4812 TDEP genes for the eight superclusters, 92.2% were TDEP for one (43.8%), two (24.7%), three (14.3%), or four (9.4%) superclusters. Fourth, we identified intersections of supercluster TDEP genes with GWAS loci, and found that TDEP genes (± 50 kb around each gene) only infrequently intersected more than one GWAS locus (7.6%). Finally and most directly, we conducted conditional analyzes to evaluate the independence of the signals across the superclusters (Fig. 2B and Supplementary Data 4–5) and across the six traits (Supplementary Fig. 7). Briefly, we found relatively consistent patterns of cross-cell-type independence: amygdala excitatory, deep-layer intratelencephalic, hippocampal CA1-3, CGE interneuron, and eccentric medium spiny neuron generally survived conditional analyzes. Superclusters shared cross-trait could reflect both common and trait-specific mechanisms. For instance, the schizophrenia relevant mechanism in the amygdala excitatory supercluster was shared with bipolar disorder but not the other four traits (Supplementary Fig. 7). Although these analyzes depend on the statistical power of the GWAS, we could identify no alternative statistical explanation or dataset redundancy to explain away the observed overlaps in Fig. 2A.

**Clinical subtyping.** MDD has the advantage of large GWAS on its clinical subtypes. For instance, SNP-heritability estimated from GWAS where cases are people with severe MDD receiving electroconvulsive therapy is greater than when estimated from GWAS where cases are identified by self-report, community, or outpatient sampling[42]. We compared the cell type enrichment between MDD subtypes viewed as clinically important (e.g., recurrent) or with empirical demonstration of greater heritability (e.g., highly severe MDD) with their counterparts[42–45], where the genetic difference between the subtypes was not due to noise[45]. Categories with any significant superclusters are presented in Fig. 2C, including recurrent MDD, MDD with functional impairment, MDD with suicidal thoughts, and postpartum MDD. In general, more signals were found in the more severe subtypes. Although the MDD subtypes largely overlap with mdd2019, there were indications of specificity; e.g., hippocampal superclusters are more related to the severe/impaired MDD subtypes and neocortical superclusters are more related to MDD with suicidal thoughts. In Supplementary Data 1 we provide the GWAS sample size and the number of genome-wide significant loci (both benchmarks of GWAS power) for all traits, and note that the MDD subtype GWAS are relatively underpowered.

Evolutionary constraint has been of considerable interest given that SNP-heritability is notably enriched for this SNP annotation[1]. Figure 2D depicts the distributions of a gene-based measure of constraint for TDEP genes for superclusters (the fraction of all CDS bases under strong constraint in 240 eutherian mammals). TDEP genes for inhibitory superclusters are more constrained than those for excitatory superclusters (in line with previous reports)[46,47]. Schizophrenia has notable enrichment in evolutionary constrained genomic loci[1,14] (Fig. 2D).

## Exome sequencing and neurodevelopmental disorders

Supplementary Data 6 presents analyzes of supercluster TDEP genes where we evaluated gene annotations derived from LD-independent methods (e.g., whole exome sequencing). As a check, we found that TDEP genes for all superclusters were significantly less likely to be "housekeeping" genes, as expected given the definition of TDEP. There were no significant associations of any supercluster TDEP genes with genes implicated via whole exome sequencing for autism or schizophrenia[12,48], but there were many associations for developmental delay and neurodevelopmental disorder (NDD)[48]. As the pattern of results was similar, we focused on NDD. NDD was significantly associated with TDEP genes for 15 of the 31 superclusters: (a) there were negative associations with Ependymal, Microglia, and Vascular superclusters (i.e., genes implicated in NDD were less likely to be TDEP genes); (b) there were eight significant associations with excitatory neuron superclusters (Amygdala excitatory, Deep-layer corticothalamic and 6b, Deep-layer intratelencephalic, Hippocampal CA1-3, Hippocampal CA4, Hippocampal dentate gyrus, Miscellaneous, and Upper-layer intratelencephalic); and (c) there were three significant associations with inhibitory neurons (Eccentric medium spiny neuron, LAMP5-LHX6 and Chandelier, and Midbrain-derived inhibitory). Notably, there was strong overlap of the NDD exome findings with the SNP-heritability enrichment for schizophrenia: of neuronal associations for NDD, 11 of 12 were also significant for schizophrenia and, of the associations with schizophrenia but not NDD, three of four were inhibitory neuronal superclusters. Clinically, a subset of people with schizophrenia have earlier NDD, and these results suggested that the two disorders may have important commonalities at a cell class level.

## Brain cell types implicated for schizophrenia

Siletti et al.[27] also identified 461 clusters of cells; superclusters contained a median of 12 clusters (IQR 8-17, ranging from 1 for Bergman glia to 92 for Splatter neurons). We conducted TDEP/S-LDSC analyzes at the cluster level for schizophrenia (Supplementary Data 7, 8). The $P$-value distribution again had modes near 0 and 1. Of the 461 clusters, 199 (43.2%) had significant (FDR < 0.05) SNP-heritability enrichment for schizophrenia. There was a strong relationship of the SNP-heritability enrichment for schizophrenia in superclusters and their component clusters with most of the significant clusters in a significant supercluster (95.0%, 189/199). Of the 10 clusters not in a significant supercluster, Splatter 403 (GABAergic cells expressing *NOS1* from amygdalar and paleocortical dissections) was exceptional (FDR 9.6e−5) and the rest had FDR values between 0.009−0.05 (eight neuronal clusters and one non-neuronal cluster with FDR = 0.04). We again find little common-variant genetic support for non-neuronal cells in schizophrenia.

In Fig. 3A, we visualized the supercluster and cluster findings for schizophrenia in the tSNE projection from Siletti et al. (their Fig. 1B). The uneven distribution of schizophrenia associations across superclusters is readily apparent. Most of the 25 strongest cluster associations (FDR < 5e-4) were from a few superclusters (Hippocampal CA1-3, Amygdala excitatory, and Eccentric medium spiny neuron; Fig. 3B). These clusters had a median of 1274 TDEP genes (IQR 1258–1286) but with modest overlap between clusters (66.2% of unique genes were TDEP for ≤ 5 clusters). Gene set analysis of the TDEP genes in these clusters highlights synaptic function, cation channels, and neuron projection (Fig. 3C, S8). Cross-trait conditional analysis at cell cluster level shows a more prominent decrease in schizophrenia-specific signal when conditioned on bipolar disorder (Fig. 3D) than conditioned on IQ (Fig. 3E). This suggests that the cell types shared by

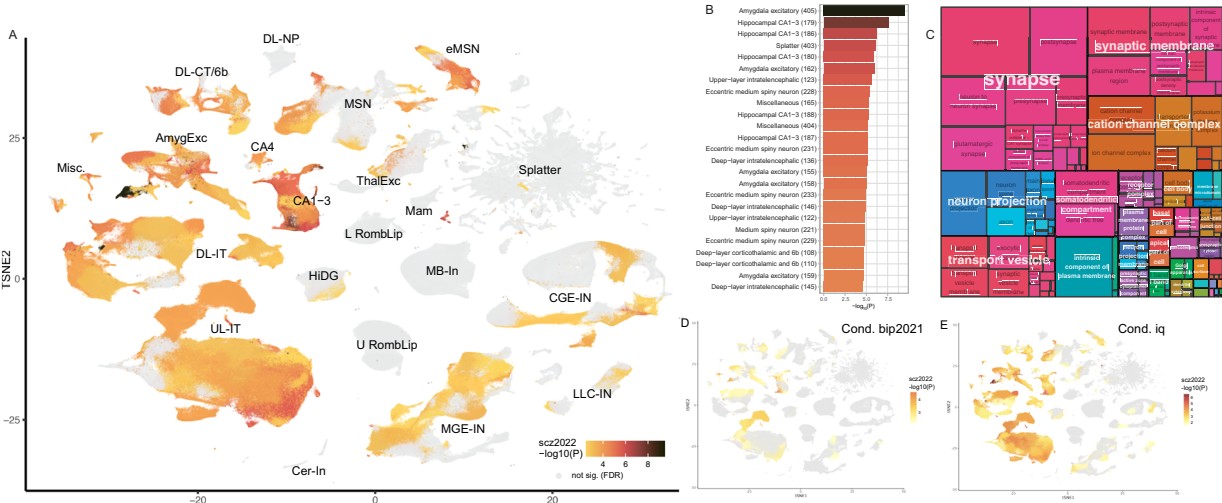

**Fig. 3 | Cluster-level SNP-heritability enrichment for schizophrenia (scz2022).**
**A** tSNE plot from Siletti et al. colored by the significance of cluster-level SNP-heritability enrichment for scz2022. One-sided *P* value was calculated based on the LDSC coefficient z-score, and we calculated FDR to account for multiple comparison. Gray indicates non-significance (FDR > 0.05). Abbreviations correspond to neuronal supercluster names in Fig. 2A. **B** Top 25 most significant clusters with enriched of scz2022 SNP-heritability out of 199 FDR significant clusters (Supplementary Data 7), same *P*-value and color definitions as in Fig. 3A. **C** Treemap plot for key GO-CC pathways in the top 25 scz2022 clusters. We evaluated gene set enrichment for the TDEP genes in each of the top 25 significant clusters, using hypergeometric test with one-sided *P*-value calculated. Significantly enriched

pathways (*P* ≤ 0.05) in all the explored clusters were integrated to highlight higher level functions. The treemap for GO-BP and GO-MF for these clusters are shown in Supplementary Fig. 7. **D**–**E** tSNE plot of clusters that remained significant for scz2022 after conditioning on bip2021 (**D**) and IQ (**E**). One-sided *P* value was calculated based on the LDSC coefficient z-score, and we calculated FDR to account for multiple comparison. The color scale and the tSNE coordinates are the same as in Fig. 3A ; gray dots are cells of non-significance clusters (FDR > 0.05). More cell types became insignificant when conditioned on bipolar disorder compared to when conditioned on IQ, suggesting the shared cell types might contribute via similar mechanisms to schizophrenia and bipolar but contribute via distinct mechanisms to schizophrenia and IQ.

schizophrenia and bipolar disorder might contribute to the two disorders through similar genetic mechanisms, whereas the cell types shared by schizophrenia and IQ might implicate distinct mechanisms.

## Analysis of anatomic regions shows distributed risk for schizophrenia risk across the brain

Connecting genetic risk to specific brain regions is important for imaging (structural or functional MRI or PET) and for identifying empirical targets for Interventional Psychiatry therapeutics (e.g., transcranial magnetic stimulation). We evaluated the distributions of cell clusters across 104 dissections (2 dissections removed, *Methods*) from 10 broad brain regions (Fig. 4A)[27]. Neuronal clusters tended to be dissection-specific whereas non-neuronal clusters were widely distributed. As we observed a lack of signal in non-neuronal cell classes for psychiatric disorders (Supplementary Data 8), and to avoid bias from different neuron/glia composition ratios, we focused on neuronal clusters. To evaluate SNP-heritability enrichment for anatomic regions, we computed the neuronal cluster proportions per anatomical dissection as weights for the cluster-level enrichments; the sum of the weighted cluster-level enrichment was the enrichment per anatomical dissection (*Methods*). First, we observed general effects in the cerebral cortex for scz2022, bip2021, educational attainment, IQ, and neuroticism (Figure 4B). Hippocampal, amygdala, and striatal (Pu and CaB) regions were also significantly enriched for the SNP-heritability of these phenotypes but with greater variability. Regional differences in hippocampal enrichment are consistent with analyzes in mouse brain[19]. Basal forebrain, thalamic, hypothalamic, cerebellum, and pons were enriched to lesser extents. To illustrate the distribution of genetic risk for schizophrenia across the brain, we depict the results using a 3D brain model (Figure 4C, E). Hippocampus and amygdala showed the highest significance of scz2022 SNP-heritability enrichment, with the top signal in the tail of hippocampus and the cortical amygdala (CoA). A detailed view of the neuronal cell type composition of the hippocampus and amygdala (Figure 4F, G) reveals that excitatory neuronal

signals were the primary contributor to the hippocampal results. For amygdala, although the highest enrichment was found in the excitatory neurons, the inhibitory neurons had greater proportions and more significant enrichments than in the hippocampus (i.e., eccentric medium spiny neuron clusters).

## Functional connectivity differences for hippocampus, amygdala, and cerebral cortex in schizophrenia

Although the prefrontal cortex is the most studied brain region in schizophrenia[49], the entire cerebral cortex had consistently significant SNP-heritability enrichment. This, together with the fact that the prefrontal cortex has extensive functional connectivity with amygdala and hippocampus[50], suggest that differences in the functional connectivity between these regions could contribute to schizophrenia mechanism. We directly assessed this hypothesis using resting-state fMRI data from individuals with schizophrenia and neurotypical controls using two independent data sets to directly replicate our results (*Methods*, Fig. 5A)[51,52]. We initially prioritized 76 brain regions that were enriched in schizophrenia SNP-heritability (FDR ≤ 0.01) (Supplementary Data 11, *Methods*).

In each data set we applied a deep neural network classifier to prioritize brain networks that distinguish cases from controls. Each set was split into five portions with balanced lengths of the time series, and we performed five folds of parallel recursive feature elimination, such that the region with the lowest contribution to the classifier was eliminated in the next iteration (Fig. 5A). In general, the data-driven networks had comparable performance to previously established, schizophrenia-relevant brain networks (i.e., the default mode and core networks, Supplementary Fig. 9A, B)[53]. Hippocampal and amygdalar regions were likely to be preserved until later runs and had significant correlation between the two replications (correlation=0.62, *p* = 0.03), suggesting their importance in distinguishing between cases and controls (Fig. 5B; Supplementary Fig. 9C, D). Connections involving any hippocampal or amygdalar regions had significantly higher feature

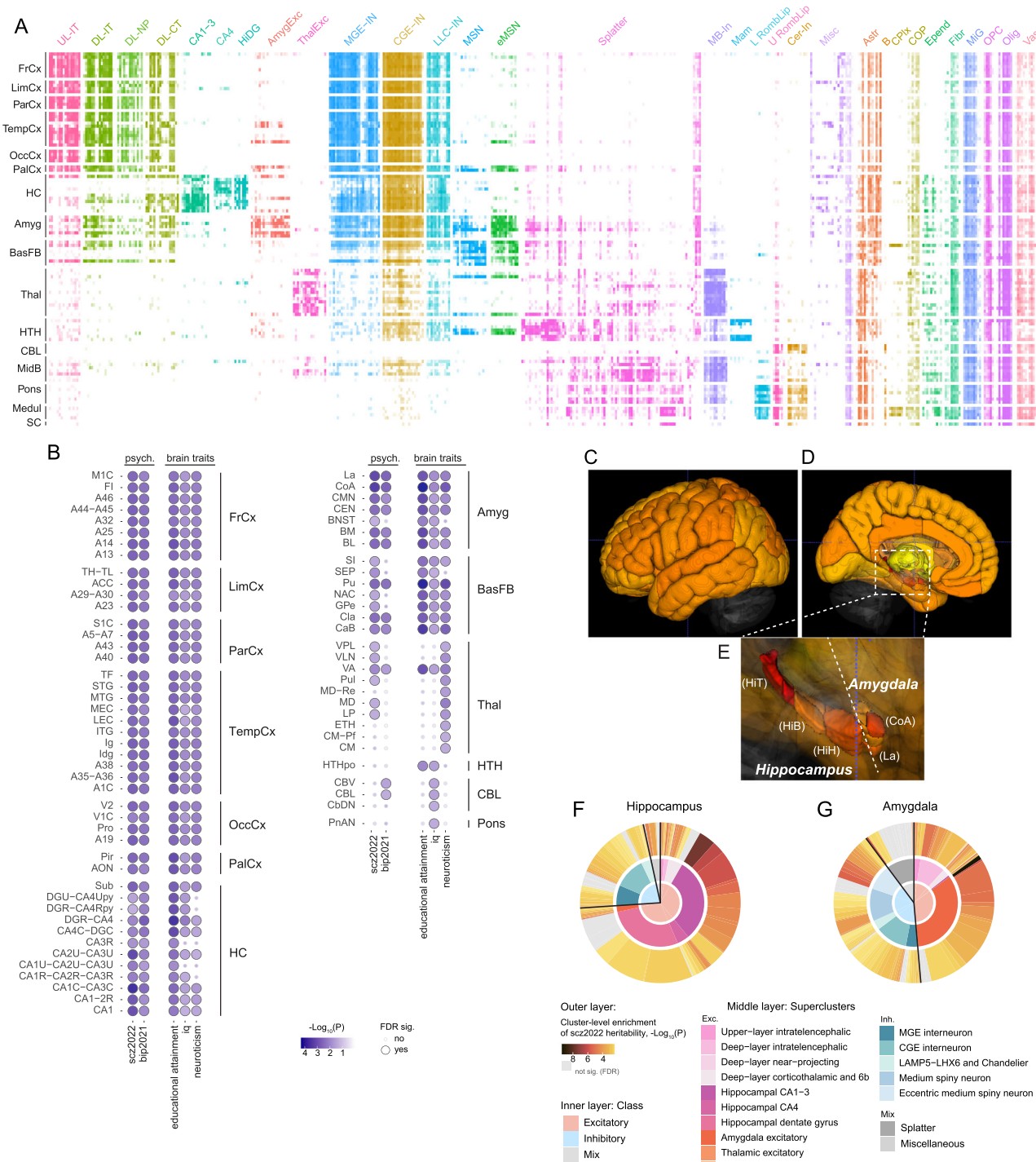

importance in both replications (Fig. 5C–F; Supplementary Fig. 9G–J). The central and lateral nuclear groups of the amygdala (CEN and La), the head and body of the hippocampus (HiH and HiB), the parietal operculum (PaO), the middle frontal gyrus (MFG), the rostral gyrus (RoG), and the anterior cingulate gyrus (CgGr) were independently confirmed being in the top 1% connections in both data sets (Supplementary Fig. 9E, F, K, L).

## Discussion

Psychiatric genomics now has empirical data strongly supporting polygenicity: multiple risk variants in "many genes" underlie the inherited tendency of these psychiatric disorders to run in families. The "genetic architectures"[7] of schizophrenia, bipolar disorder, MDD, and other major psychiatric disorders – causes of considerable human suffering – are dominated by large numbers of common genetic variants of small effect[14,54,55]. The neurobiological implications of these secure and replicated genome findings are, however, unclear. In this paper, we rigorously evaluated the hypothesis that the accumulated findings implicate physically identifiable brain structures (i.e., cell types and anatomical regions). By necessity, our prior work was based on mouse brain atlases[17,18] and here we extend our work using a detailed and comprehensive human brain atlas[27]. Using fMRI as an orthogonal modality, we show that a data-driven model trained to classify schizophrenia cases from neurotypical controls prioritizes connections involving subcortical structures, particularly the amygdala and hippocampus. These findings provide support for

**Fig. 4 | Leveraging cluster-level SNP-heritability enrichment to brain regions.**
**A** Heatmap of the scaled proportion of each cluster in each dissection. X-axis are clusters grouped and colored by superclusters. Abbreviations of supercluster names see Fig. 2A. Y-axis are brain dissections grouped to broader regions. Abbreviations of regions: FrCx: frontal cortex, LimCx: limbic cortex, ParCx: parietal cortex, TempCx: temporal cortex, OccCx: occipital cortex, PalCx: paleocortex, HC: hippocampus, Amyg: amygdala, BasFB: basal forebrain, Thal: thalamus, HTH: hypothalamus, CBL: cerebellum, MidB: midbrain, Medul: medulla, SC: spinal cord. Nomenclature follows Siletti et al.[27]. Each cell represents the scaled proportion of a cluster in a dissection (Methods and Supplementary Data 9). **B** Significance of SNP-heritability enrichment for anatomical dissections, where one-sided $P$-value was derived from cluster-level significance weighted by the proportion of clusters per dissection (Methods). Dot color indicates significance reflected by -log10(P) with darker blues indicating greater significance; dot size indicates the significance after adjusting for multiple comparisons across dissections for each trait at FDR ≤ 0.05. Only phenotypes and brain regions with any significant signal are shown (full

results in Supplementary Data 10). **C–E** Anatomic dissection results of scz2022 plotted on a 3D brain model (C-lateral view, D-sagittal view, E-enlargement of hippocampus and amygdala; visualizing the 3D Allen Brain model in ITK_SNAP v3.8.0, see "Methods"). Red indicates greater and yellow lesser significance at FDR ≤ 0.05, and gray and transparent indicates non-significance. Unsampled cerebral cortical regions are colored per the sampled regions (as mean of the enrichment Z-scores for sampled cerebral cortical regions). HiH: head of the hippocampus; HiB: body of the hippocampus; HiT: tail of the hippocampus; CoA: anterior cortical nucleus of the amygdala; La: lateral nucleus of the amygdala. **F–G** Greater detail for hippocampus and amygdala. The outer layer indicates clusters; the size is the proportion of the cluster, and the color indicates cluster-level significance of scz2022 SNP-heritability enrichment (color scale and the one-sided $P$-value are the same as in Fig. 3A). The middle layer is colored by superclusters, and the inner layer is colored by classes. Splatter and Miscellaneous have both excitatory and inhibitory components and were categorized as "Mix".

---

convergence between different data-driven approaches on the brain regions identified through our genomic analysis.

## Human brain cell data with regional resolution

Consistent with previous reports[17,18], neuronal cell types had substantially increased SNP-heritability for psychiatric disorders (schizophrenia, bipolar disorder, and major depression) and brain traits (educational attainment, iq, neuroticism, insomnia, alcohol consumption, and smoking initiation). The anatomical data allowed detection of trait-relevant brain regions. Regional signals were distributed across the cerebral cortex and subcortical cerebral nuclei. While confirming previous results based on mouse scRNA-seq data for hippocampal and neocortical excitatory neurons[17,18], based on human data we have identified more relevant cell types, such as amygdala excitatory neurons, which were the most significantly enriched cell type in the entire brain as well as subcortical projecting GABAergic neurons for schizophrenia, which were undistinguishable in previous mouse datasets. Our study highlights neocortical interneurons derived from caudal ganglionic eminence which mainly contact other interneurons rather than interneurons expressing somatostatin or parvalbumin (although alterations of both of the latter have been reported in schizophrenia cases[56,57]).

## Cross-cell-type and cross-trait findings

We observed broad involvement of brain regions in several psychiatric disorders and brain traits; at the same time, each phenotype had multiple supercluster-level signals. The signals were largely statistically independent between the superclusters (Fig. 2B, Supplementary Data 4, 5), suggesting cell type specific mechanisms contributing to the same trait. Meanwhile, the same cell types shared between traits might contribute to the traits through similar (e.g., schizophrenia and bipolar disorder, Fig. 3D, S7) and also distinct mechanisms (e.g., schizophrenia and IQ, Fig. 3E, S7). Interestingly, genes implicated by exome sequencing in neurodevelopmental disorders largely pointed at the same brain cell types. Combined with our analyzes of TDEP genes, GWAS loci, and conditional analyzes, we believe that these results support cell types as contributing to phenotypically diverse traits. This suggests convergence, that these clinically distinctive phenotypes are rooted in both shared and different functional aspects of the same brain cell types.

With the available GWAS for MDD subtypes, we were able to infer important cell types for clinical subtypes. We observed that more superclusters were implicated for severer subtypes of MDD. It is possible that more severe subtypes convey higher genetic risk and therefore greater statistical power in the GWAS. It is also possible that the different subtypes had partially distinct etiologies, as suggested by imperfect genetic correlations among subtypes[45,58]. More precise interpretations of the cell types can be made when larger subtype GWAS become available.

At the level of brain regions, the results pointed out the importance of subcortical structures, especially the hippocampus and amygdala, underlying the mechanisms of pathological (e.g., schizophrenia) and healthy (e.g., educational attainment) phenotypes. The results in amygdala are in agreement with other findings that implicates changes in its structure in psychiatric disorders[59–61]. From a clinical perspective, amygdalar dysfunction agrees with decreased ability to ascribe correct valence and attention to sensory inputs[54].

## Implications for schizophrenia

Neocortical regions presented similar enrichments across the brain even though certain neocortical regions have been implicated in psychiatric disorders (e.g., dorsolateral prefrontal cortex and schizophrenia[49]). This is likely explained by the similar cell type composition across the neocortical regions[62], and highlights the importance of functional connectivity in the underlying mechanisms of schizophrenia[63]. The TDEP genes of the top scz2022 clusters highlighted synaptic functions and neuronal projection suggesting mechanistic connectivity between cells.

Our findings from independent fMRI connectivity data sets confirmed the importance of cortical regions including the middle frontal gyrus (BA 46 in the snRNA-seq dataset), rostral gyrus (BA 32, snRNA-seq dataset), anterior cingulate gyrus, and the parietal operculum, but also further highlighted the differential connectivity of the amygdalar and hippocampal regions between schizophrenia cases and controls. Hippocampus and amygdala are both involved in emotional memory processing, and a directed influence from the amygdala on the hippocampus has been suggested during fear processing in response to emotionally salient information[64,65]. Our method thus suggests plausible brain regions for schizophrenia etiology and calls for further investigations into these areas and their functional connectivity, which may hold new candidates for modulation using non-invasive therapeutics.

These results need to be considered with limitations (see also the Supplement of reference [66]). First, brain regions were not equally sampled (Supplementary Data 12), despite the snRNA-seq dataset having the most comprehensive coverage of the adult human brain to date, and we cannot rule out enrichments of trait heritabilities in other brain regions. Second, the Human Brain Atlas is from a few adults and does not capture variability between neurotypical individuals or individuals with severe and enduring mental disorders or variability across the lifespan (especially during brain development). Finally, important limitations need to be considered when interpreting the results from the fMRI analysis. The molecular mechanism behind the fMRI blood oxygen level-dependent (BOLD) signals is not fully understood, but recent technological advancements in measuring local neural activity, neurovascular responses, and spatial transcriptomics will likely provide deeper insights[67,68]. Our interpretation of the converging findings of the

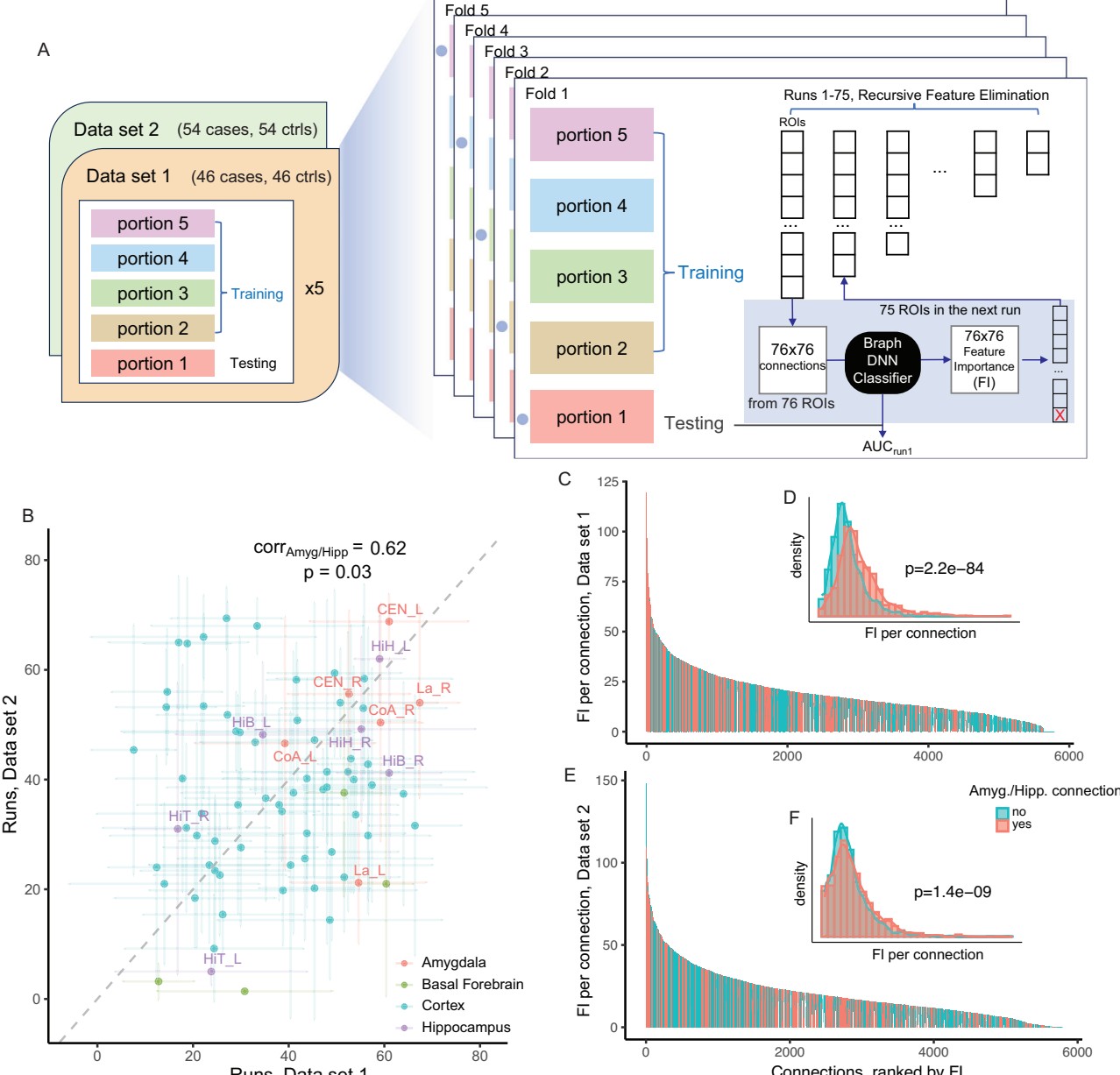

**Fig. 5 | Functional connectivity networks derived from brain regions enriched for schizophrenia SNP-heritability distinguished cases and controls.**
**A** Workflow of the fMRI analysis in two independent data sets, detailed in "Method" section. In each fMRI data set, five folds of recursive feature eliminations were performed using the training set. In each run, the model (network) was evaluated in the independent testing set using the area under curve (AUC) of the receiver operator characteristic curve (ROC). **B** Number of runs for which each region was preserved in the recursive feature elimination in fMRI data set 1 (46 cases and 46 control) and data set 2 (54 cases and 54 controls). The dot indicates the mean runs in each data set for the region (with standard deviation bars per data set); the color indicates the broader areas each region belongs to. The Pearson's product-moment correlation between the runs in the two data sets was 0.02 ($p = 0.90$) for all ROIs,

but 0.62 ($p = 0.03$) for the amygdalar and hippocampal regions, and these regions were also more likely to be preserved to later runs (also Fig S9 C, D). (C) Pairwise connections between regions ranked by the feature importance (FI, $y$-axis) per connection in data set 1. The color indicates connections involving any amygdalar or hippocampal regions (red) or not (teal). **D** Density plot of the FI in amygdalar or hippocampal connections (red) and other connections (teal) in data set 1. The FI of amygdalar or hippocampal connections are significantly higher than the others (two-sided Welch two sample t-test, $p = 2.2\text{e-}84$). **E** Connections ranked by FI in data set 2. **F** Density plot of the FI in data set 2. The FI of amygdalar or hippocampal connections are significantly higher than the others (two-sided Welch two sample t-test, $p = 1.4\text{e-}9$).

importance of brain regions from genomic and fMRI analyzes does not imply a mechanistic link between the transcriptomic changes in neurons themselves and changes in local BOLD signal, but rather emphasizes changes in functional connectivity (i.e., coordination) between the brain areas which perhaps suggest changes in synaptic function/targeting. This interpretation is supported by the finding that schizophrenia genetics implicates synaptic function[14]. Furthermore, the sample size of

the two independent fMRI data sets was limited given the case-control setting, and thus our initial results will have to be confirmed in larger data sets as they emerge. In addition, the stereotactic mapping of the sampled brain regions in the transcriptomic atlas to the 3D brain model was based on the detailed region information in the atlas and, we were unable to account for bilaterality given that snRNA-seq data were from the right hemisphere and the fMRI data were bilateral.

In conclusion, our findings extend prior work by showing the human brain localization of genomic regions implicated in three psychiatric disorders, three relevant brain traits, and in genes implicated in neurodevelopmental disorders. The findings point at largely overlapping cell types and brain regions (albeit different subsets of genes). These findings provide a framework for understanding the polygenicity of complex psychiatric disorders and brain traits as well as suggesting hypotheses for future research, such as the transcriptomic differences in subcortical regions for the mechanisms of severe psychiatric disorders. Our findings underscore the value of single-cell transcriptomics in decoding the polygenicity of psychiatric disorders and provide the hope that the genomic, transcriptomic, and brain imaging modalities can be integrated to offer a richer understanding toward common biological targets.

## Methods

### Human Brain Atlas single-nucleus RNA-seq (snRNAseq)

We used the Human Brain Atlas snRNAseq data set from Siletti et al.[27]. This atlas consists of 3.369 million nuclei successfully sequenced using snRNAseq. The nuclei were from adult postmortem donors, and the dissections focused on 106 anatomical locations within 10 brain regions. Following quality control, the nuclear gene expression patterns allowed the identification of a hierarchy of cell types that were organized into 31 superclusters and 461 clusters. In the current paper we use the same naming system for the cell types and the brain regions as in Siletti et al.

### Genome reference and gene models

The reference genome and gene models were with respect to a modified version[27] of the GENCODE primary assembly (GRCh38.p13, v35, 3/2020, hg38)[69]. As hg19 is typically used by GWAS, we also obtained GRCh37/hg19 gene coordinates from GENCODE (v35). In these analyzes, we focused on 18,090 genes with these characteristics: protein-coding, mapped to canonical autosomes (chr1 to chr22), not in the extended major histocompatibility (MHC) region (chr6:25-34 mb), and expressed in ≥ 1 of the 461 cell clusters. Explanations for these choices follow.

### Protein-coding biotype.

The modified GENCODE assembly used by Siletti et al.[27] contained N = 51,263 genes with TPM > 1 in one or more cluster cell types. In GENCODE, these genes are grouped into 30 biotypes ranging from rare ("scRNA" and "vault_RNA") to common ("protein_coding" (N = 19,153) and lncRNA (16,021)). Siletti et al. used the 10X Genomics Chromium Next GEM Single Cell 3' Reagent Kits (v3) whose beads contain a 30 nt poly-dT tail and thus will most consistently capture 3' poly-adenylated RNA transcripts (in humans, these include mature protein-coding and lncRNA transcripts). For each biotype, we summed the number of occurrences of any gene with TPM > 1 over all superclusters and found that only protein-coding and lncRNA genes had appreciable transcript detection. For instance, 20 biotypes had < 100 detected transcripts and 28 had < 8600 detected transcripts in any supercluster. We chose to drop lncRNA genes and only include protein-coding genes. First, although a small number of lncRNAs have been shown to have biological functions, the annotation of most lncRNAs is currently unknown. In these data, 81.4% of the lncRNAs had a generic annotation (e.g., "novel transcript"). Second, the lncRNA were not strongly expressed and/or were not well-captured by the 10X Genomics kit: the largest median expression of lncRNAs in the superclusters was only 0.20 TPM (compared to 11.5 TPM for protein-coding genes).

### MHC.

The extended MHC (eMHC) is the largest block (~8 mb) of high linkage disequilibrium (LD) in the genome (excluding pericentromeric regions)[70]. For instance, of the 23,731 significant SNP associations with schizophrenia, 4527 (19.1%) are in the eMHC region[14]. These generally correspond to highly correlated genetic variants. We removed GWAS SNPs and snRNAseq data in the eMHC as in our prior papers and

as recommended by the S-LDSC authors[17,18,71]. However, to evaluate the impact of this choice, we recalculated the TDEP estimates while including protein-coding eMHC expression data (N = 259), and found that a small number of eMHC genes had a TDEP flag in superclusters (median 10 genes, IQR 8–13). As the median number of TDEP genes per supercluster was 1287, the potential eMCH region contribution to a TDEP list is 0.8% (10/1287). The impact is likely not consequential.

### Autosome.

Sex chromosome genes were removed. chrY is rarely included in GWAS; in a recent build of the NHGRI/EBI GWAS Catalog[72], there were only 5 significant SNP associations to any trait whereas a similarly sized chromosome (chr22) had > 3000 GWAS hits. In addition, chrX data are inconsistently included in the summary statistics from GWAS papers[73], and are under-represented in the GWAS catalog: chrX has 1149 hits whereas the similarly sized chr7 and chr8 have 9438 and 9745 associations. In a sense, the choice to exclude sex chromosomes was made for us as, for the GWAS traits we analyzed (Supplementary Data 1), none had chrY and a minority had chrX results. To evaluate the impact of this choice, we recalculated the TDEP estimates while including protein-coding chrX expression data (N = 779), and found that some chrX genes had a TDEP flag in superclusters (median 49 genes, IQR 41–60). As the median number of TDEP genes per supercluster was 1287, 49 genes (3.8%) may have had a small impact. We are unable to address this issue given the data available, and this is unquestionably a topic for future research.

### GWAS summary statistics

We conducted multiple searches to identify potential GWAS (i.e., PubMed, Psychiatric Genomics Consortium downloads page, NHGRI/EBI GWAS catalog). We previously have shown the importance of genetic architecture on the informativeness of our approach (see references [14,43]. The number of loci (genomic regions harboring multiple correlated genome-wide significant SNPs, defined below) is particularly important. We required > 10 loci for inclusion (with a few intentional exceptions). Supplementary Data 1 summarizes the GWAS included in our analyzes. These 36 primary GWAS are the largest studies per trait that we could obtain as of 4/2023 and whose use was compatible with our publication strategy (some prepublication GWAS required submission delays and others were not freely available, e.g., 23andMe).

- We included five psychiatric disorders (ADHD, bipolar disorder, major depressive disorder, problematic alcohol use, and schizophrenia). We did not include multiple important psychiatric disorders due to low numbers of loci (e.g., anorexia nervosa, autism).
- We included eight neurological diseases: Alzheimer's disease, amyotrophic lateral sclerosis, epilepsy, hearing loss, migraine, multiple sclerosis, Parkinson's disease, and stroke. For Parkinson's disease, we used the results of Nalls et al. 2019 excluding 23andMe samples[74].
- We included nine structural brain MRI measurements: brainstem volume, caudate volume, neocortical surface area, and putamen volume. Because these MRI measures describe important brain features (and often the anatomic regions from Siletti et al.), we also included accumbens volume, amygdala volume, neocortical thickness, pallidum volume, and thalamus volume.
- We included nine trans-diagnostic brain traits of clinical salience (alcohol use, smoking traits, and insomnia) or which may be clinical stratifiers (educational attainment, IQ, neuroticism). Suicide phenotypes were included due to their importance in the current mental health crisis.
- Finally, we selected five control traits with large numbers of loci but whose genetic architectures are not rooted in the central nervous system: height, body mass index, hematocrit, lymphocyte count, and neutrophil count.

For MDD, we included 19 additional GWAS to assess within-disorder questions (no requirement for minimum number of associations; Supplementary Data 1). As etiological heterogeneity is likely for depressive disorders, we evaluated whether heterogeneity was associated with different brain cellular enrichments. We focused on the clinical contexts in which a major depressive episode (MDE) can occur. The classical delineation of MDE is in the context of unipolar or bipolar disorder. An MDE can occur as major depressive disorder (MDD, unipolar MDE with no history of mania or hypomania), MDE with a history of mania (bipolar disorder type 1), and MDE with a history of hypomania (bipolar disorder type 2). These conditions have different genetic correlations with bipolar type 2 being more similar to MDD and bipolar type 1 being more similar to schizophrenia[54]. MDD can occur in different ways clinically and across the lifespan. We evaluated MDD subtypes viewed as clinically important (degree of severity, typical vs atypical symptom pattern, with or without comorbid anxiety disorder) or with empirical demonstration of greater heritability: highly severe MDD (people receiving electroconvulsive therapy for MDE), early-onset MDD, recurrent MDD, and postpartum depression[42–45].

**Processing and quality control (QC).** After we obtained GWAS results from the primary sources, we conducted range checks for logistic or multiple regression betas, standard errors, and *P*-values (removing SNPs with highly unlikely values). We then processed all sumstats using the cleansumstats pipeline (https://github.com/BioPsyk/cleansumstats):

- We determined genome build by comparing SNP positions to dbSNP (build 151)[75].
- Using UCSC::liftOver, we ensured we had sumstats in hg38/GRCh38 and hg19/GRCh37 coordinates (GWAS tend to use hg19 and genome annotations tend to use hg38).
- We removed insertion/deletion polymorphisms, duplicate entries, and chromosomal locations not in [chr1-chr22] and noting that sex chromosome data are inconsistently included in GWAS summary statistics[73].
- We required that each variant match dbSNP (build 151) by rsID and that the GWAS sumstats SNP alleles (effect/other allele) matched REF/ALT in dbSNP (flipping to + strand if required).
- We removed homozygous/monomorphic SNPs, SNPs with alleles not in [ACGT], and strand-ambiguous SNPs (A/T or C/G; these are also removed in LD score regression).
- Given our use of S-LDSC (below, and as typically done), we excluded the extended MHC region (chr6:25-34 mb) due to its exceptionally high LD.

**Genomic loci.** We used the clumping algorithm in plink[76] to identify loci for the GWAS included in this report. The LD reference was the European subset of the 1000 Genomes Project (phase 3)[77] with parameters: p1 = 5e-8, p2 = 5e-6, $r^2$ = 0.1, and window size of 3000 kb. Overlapping loci and loci within 50 kb of each other were merged.

**Description of the primary GWAS.** We then conducted basic checks including the number of SNPs after QC, the number of genome-wide significant SNPs (*P* < 5e-8, after QC), inflation statistics ($\lambda$ and LDSC intercept), and SNP-heritability (Supplementary Data 1). For the primary GWAS traits, the numbers of significant loci were positively correlated with sample size (Spearman *p* = 0.62, *P* = 4.9e-5) and the number of cases (binary traits, Spearman *p* = 0.73, P = 0.0012). Supplementary Fig. 1 illustrates some key features of the GWAS included in the primary analyzes.

Supplementary Fig. 2 provides more data about the primary GWAS traits. The SNP-heritability estimates on the diagonal are consistent with the primary reports (any differences are due to our use of sample subsets like European subjects or after removing 23andMe results). The off-diagonal elements show the interrelationships of the primary GWAS traits via a heatmap of genetic correlations ($r_g$ from LDSC). The pattern of genetic correlations are consistent with prior reports[3]. In general, we note: (a) positive intercorrelations for psychiatric disorders and brain traits, (b) transdiagnostic negative correlations of educational attainment and IQ with multiple conditions, (c) relatively weak correlations for neurological diseases, and (d) isolated correlations for structural MRI measures.

## Relative versus absolute gene expression

We use top decile of expression proportion (TDEP) to identify genes whose expression typifies each supercluster (-1300 genes per supercluster). Li et al.[35] determined that S-LDSC with TDEP had power and false positive rates that, jointly, were equivalent or superior to 8 other methods. See the *Statistical analysis* section below for definition of TDEP, TPM, and the Li et al. results[35].

Here, we compare relative vs absolute measures of gene expression. TDEP is a relative measure, the expression of a gene in one cell type divided by the total expression across all cell types. In contrast, TPM (snRNAseq count data in a cell type normalized to molecule transcripts per million) is more of an absolute method that reflects the number of RNA molecules in specific cells. We thus contrasted TDEP and TPM.

First, as a basic data visualization, Supplementary Fig. 3 depicts the relation between TDEP and TPM for each of the 31 superclusters. For most superclusters, gene expression was greater in TDEP genes. This was particularly notable for non-neuronal superclusters where the median expression was far higher for TDEP genes (e.g., oligodendrocyte median 102.1 vs 18.9 TPM in TDEP genes vs all other genes). The excitatory neuronal superclusters had similar appearances in Supplementary Fig. 3 except for upper rhombic lip and lower rhombic lip being somewhat different. Inhibitory neuronal superclusters appeared relatively similar.

Second, we created two data matrices; rows were 18,090 autosomal, protein-coding genes, columns were 31 supercluster classes, and the elements were either $\log_2(\text{TPM} + 1)$ or TDEP (1 = yes, 0 = no). The TPM matrix is obviously far more nuanced and detailed than the TDEP version. The results of UMAP/HDBSCAN are depicted in Supplementary Fig. 4. In both instances, the high-dimensional data could be visualized as 3 distinct clusters. Cluster positions are arbitrary but the solutions are otherwise qualitatively similar, clusters containing: (a) all non-neuronal cells; (b) all neocortical excitatory neurons plus medium spiny neurons; and (c) all inhibitory neurons and non-cortical excitatory neurons.

This is notable because TDEP faithfully recapitulates the multivariate structure of the supercluster data based on the more information-rich and full gene expression matrix based on TPM. The TDEP 0/1 flags efficiently capture the high-dimensional density structure of the TPM expression matrix.

Third, we contrasted gene set analyzes using GO[37]. The GO gene set analyzes were based on TDEP genes and separately for top decile TPM. As both variables are defined by the deciles and coded TRUE/FALSE, similar numbers of genes are being compared. The background was 18,090 autosomal, protein-coding genes.

These analyzes yielded 322,462 comparisons (31 superclusters x 10,422 GO sets). The correlation in hypergeometric *P*-values for TDEP with top decile TPM was modest (Spearman *p* = 0.414). For significance at FDR < 0.05, TDEP was more conservative than top decile TPM in implicating GO gene sets (3.27% vs 8.40%). Of all pathways, the two methods agreed for 92.2% (both non-significant for 291,094 or 90.30%, and both significant for 6228 or 1.93%). There were fewer disagreements for TDEP == TRUE and top decile TPM == FALSE (4329 or 1.34%) than the reverse (TDEP == FALSE and top decile TPM == TRUE, 20,811 or 6.45%). Checks of disagreements with top decile TPM FDR < 0.05 and TDEP FDR > 0.5 (larger FDR applied to avoid edge cases) revealed some confusing results: e.g., synaptic genes sets with non-neuronal supercluster classes including astrocyte, Berman glia, oligodendrocyte, fibroblast, and vascular. These disagreements

tended to be the same (i.e., about half of these pathways were implicated in ≥5 superclusters). We believe that the differences were strongly influenced by broadly (and often highly) expressed "housekeeping genes" that are prevalent in top decile TPM but not in TDEP (by definition). The top decile TPM gene set findings are in contrast to those for TDEP (presented in Supplementary Data 2) that captured the expected (if not canonical) biological processes, cellular compartments, and molecular function of the 31 superclusters.

Taken together, these results support our use of TDEP as a means to identify genes that are enriched for biological processes related to the cellular identity and specific function of superclusters.

**TDEP in human and mouse brain studies.** Our prior papers were based on scRNA-seq mouse neural surveys[17,18] with the key limitation of a necessary reliance on protein-coding genes with a high confidence, 1:1 mouse-human ortholog. Of the 18,090 genes we evaluated (autosomal, protein-coding, not in eMHC, TPM > 1 in ≥ 1 cluster), 14,398 (79.7%) had a high confidence, 1:1 mouse-human ortholog. As a sanity check, we compared TDEP genes for 23 mouse brain cell types used in Skene et al.[66] to the 31 Human Brain Atlas supercluster using hypergeometric gene set analysis. There was considerable consistency across these datasets despite different technologies and organisms. For example, there was the greatest overlap of: mouse "pyramidal CA1" with human hippocampal CA1-3 (fdr = 1e−140); mouse "pyramidal somatosensory" with human upper-layer intratelencephalic (fdr = 1e−134); mouse "oligodendrocyte" with human oligodendrocyte (fdr = 1e−183); and mouse "endothelial mural" with human vascular (fdr < 2e−208). As expected, the mouse signal for some cell types resolved into more precise human superclusters: mouse "interneurons" was associated with four human inhibitory neurons and the three mouse hypothalamic cell types contained human ependymal as well as excitatory and inhibitory neuronal TDEP genes.

**Summary.** Thus, we believe that TDEP is a defensible choice. Its relative nature can be a limitation in extreme instances, but it is a principled and intentional choice that we evaluated extensively in this section. Further support can be found in method comparison studies: see discussion of Li et al.[35] in the section titled "Choice of TDEP/S-LDSC". The similarities in Supplementary Fig. 4 are reassuring and TDEP's more conservative and the face-valid gene set results strengthen its appeal.

### Properties of Human Brain Atlas superclusters

We use TDEP to identify genes whose expression typifies each supercluster (-1300 genes per supercluster). Multiple choices that we made in using TDEP are explained above, and the *Statistical Analysis* section below provides definitions and further justification. We posit that TDEP genes for a cell type are enriched for biological processes related to cellular identity and function, and we evaluated this assumption in multiple ways.

**Traditional classification and gene set analysis.** Supplementary Data 2 characterizes the 31 supercluster cell classes: 10 non-neuronal and 21 neuronal cell classes (13 excitatory, 7 inhibitory, and 1 mixed neuronal class). Non-neuronal cell classes generally derived from dissections across the brain: e.g., astrocyte, ependymal, fibroblast, oligodendrocyte, and vascular cells were identified in many anatomical regions (with the exceptions of Bergmann glia and choroid plexus). Neuronal cell classes usually had a predominant anatomical region: e.g., deep-layer near-projecting and upper-layer intratelencephalic excitatory neurons from neocortical dissections and the 3 hippocampal cell classes were from hippocampus (the main exceptions were the miscellaneous and splatter).

Supplementary Data 2 contains Gene Ontology (GO) gene set analysis for TDEP genes[37]. Results for non-neuronal cells suggested a markedly diverse range of significant GO terms that were consistent with the supercluster labels: astrocyte/biological adhesion, choroid plexus/cilium, microglia/immune response, oligodendrocyte/ensheathment of neurons, oligodendrocyte precursor/gliogenesis, and vascular/vasculature development. In contrast, for most neuronal cell classes, GO terms focused directly on synaptic biology.

A small set of genes (215, 1.19%) had TDEP in 10–14 supercluster classes. These genes contained multiple cadherins, calcium channel subunits, muscarinic receptors, GABA receptors, glutamate ionotropic and metabotropic receptors, potassium channel subunits, sodium channel subunits, synaptotagmins, and transmembrane proteins. Despite a small number of genes that usually limits gene set analysis, these 215 genes were enriched for 38 SynGO[78] synaptic cellular compartment and biological process annotations (e.g., presynapse $P_{hyper} = 3.9e{-}11$ and postsynapse $P_{hyper} = 4.1e{-}11$).

We also addressed the inverse question, the 21.0% of genes that were not in a TDEP gene list for any supercluster. These genes were highly enriched for: (a) genes expressed at high and consistent levels across tissues ($P_{hyper} < 2.2e{-}308$, a definition of "housekeeping" genes)[36]; (b) evolutionarily constrained genes ($P_{hyper} = 1.6e{-}108$)[1]; (c) a range of GO biological process annotations pertaining to RNA processing, gene regulation, and cellular energetics ($P_{hyper} < 1e{-}40$); and (d) notably, no synaptic processes ($P_{hyper} = 1$)[78]. As expected, genes whose supercluster expression are non-specific were dominated by fundamental processes common to most cells and which tend to be highly constrained in placental mammals.

**Gene expression.** The Human Brain Atlas data[27] consist of snRNAseq on 3.369 million nuclei from adult postmortem donors and 106 anatomical locations within 10 brain regions that were then organized into 31 supercluster classes. We made a data frame with columns for the Ensembl gene identifier and each of the 31 superclusters along with 18,090 rows (for each autosomal, protein-coding gene expressed in ≥ 1 cluster). The elements are the expression of a gene in each cell class (as TPM, molecule transcripts per million). Supplementary Fig. 3 shows the relation between TPM and EP by supercluster class. At this level of analysis, there is considerable diversity in terms of the gene repertoire and expression level. Many of these genes will be responsible for core physiological processes and are robustly expressed in most cells (e.g., "housekeeping" genes).

**Genomic location and supercluster TDEP genes.** Genes are not randomly positioned in the human genome but rather show a marked tendency to occur in clumps. For instance, we can divide the genome into a regular set of 100 kb bins. After removing bins that were entirely composed of "N" (unknown) bases and after excluding chrX, chrY, and the extended MHC region (as noted above), there were 27,597 × 100 kb bins. We assigned the transcription start site (TSS) of 18,090 protein coding genes to these 100 kb bins, and tabulated the observed number of bins with 0, 1, 2, ... TSS. We created an expectation using random sampling with replacement. In the human genome, we observed that 97.1% of all 100 kb bins have no TSS (i.e., 2.9% contain from 1-14 TSS). The observation is markedly different from the random expectation: the fraction of bins without a TSS ranged from 78.3–80.0% (1000 trials). As the observed fraction of bins with no TSS (97.1%) was never approached, this implies that an empirical probability of this observation is far less than 0.001.

If protein-coding genes clump or cluster together in the genome, then TDEP genes are likely to cluster as well, given that they are a subset of all protein-coding genes. We thus assessed whether TDEP genes co-occurred in excess of the fundamental clumping of protein-coding genes. For each of the 10,027 × 100 kb bins containing a TSS for a protein-coding gene, we tabulated the total number of protein-coding TSS (nTss) and the number of these that were TDEP genes (separately for each supercluster). We fit 31 linear regression models

(nTdep$_i$ ~ nTss) and saved the studentized residuals (i.e., transformed to mean 0 and standard deviation 1). The studentized residuals had minimum values > −3 and the 75th percentiles were around −0.2. However, for ~14% of the bins, there were far more TDEP TSS than expected given the number of TSS (defined as studentized residuals > 3). To visualize these relationships, we computed the Spearman correlations for the studentized residuals of 31 supercluster cell classes and depicted the correlation matrix as a heatmap following hierarchical clustering (Supplementary Fig. 5). Note that: (a) non-neuronal supercluster classes tend to correlate, specifically ependymal-choroid plexus, Bergman glia-astrocyte, oligodendrocytes, and vascular-fibroblast; and (b) neuronal cells classes clump in 2 groupings.

The groupings in Supplementary Fig. 5 are strongly reminiscent of those in Supplementary Fig. 4. We believe that this is notable given that the input data come from different sources (the latter from genomic location and the former from gene expression). This observation implies a role for co-expression of genes in genomic regions and supercluster identity.

### Statistical analysis for SNP-heritability enrichment

**Gene expression specificity/expression proportion.** We calculated gene expression specificity per cell type as expression proportions (EP). The following steps were done for cell types at the supercluster level (N = 31) and then at the cluster level (N = 461). For each cell type, we normalized the Siletti et al.[27] snRNAseq count data to molecule transcripts per million (TPM, equation I). We then computed EP per cell type as the normalized expression divided by the sum of normalized expression across cell types for each gene (equation II).

$$Exp_{g,c} = \frac{Raw_{g,c} \times 1e6}{\sum_g Raw_{g,c}} \qquad (1)$$

$$EP_{g,c} = \frac{Exp_{g,c}}{\sum_c Exp_{g,c}} \qquad (2)$$

As in our prior papers, we selected genes in the top decile of expression proportion (TDEP) per cell type with normalized expression > 1 TPM.

**SNP-heritability enrichment in cell types.** Stratified LD score regression (S-LDSC) is widely used to evaluate whether a specific genome annotation is enriched for GWAS findings that contribute a greater proportion of SNP-heritability (also known as SNP-based heritability) to the common variant genetic architecture. It incorporates an empirical approach to LD correction via LD scores (the sums of local $r^2$ LD values) and includes multiple other genome features to increase model stability[71,79]. We used S-LDSC[71] to evaluate whether the set of ~1300 genes with top decile EP for each supercluster-level cell type (N = 31) or cluster-level cell type (N = 461) had significant SNP-heritability enrichment. Enrichment is calculated for the SNP-set relative to a null hypothesis that all SNP contribute equally to the SNP-heritability. Gene boundaries were expanded by ± 100 kb. S-LDSC was run for each combination of GWAS summary statistics and cell type (at supercluster- and cluster-levels). We provide additional justification for these methodological choices below. As recommended, enrichment $P$-values were computed from the "Coefficient_z-score"[79]. For each GWAS trait, we adjusted for multiple comparisons using false discovery rate (FDR) using R::rstatix::adjust_pvalue(method = "fdr").

**Choice of TDEP/S-LDSC.** Multiple groups have proposed algorithms by which to connect GWAS results to specific cell types. In addition to top-decile EP (TDEP)/S-LDSC, published methods include (in alphabetical order): CELLEX, DIALOGUE, EPIC, EWCE, MAGMA, RolyPoly, sc-linker, and scDRS[19–26]. These methods evaluate the association of

GWAS signals with gene expression specificity in a given cell type as measured by single-cell or single-nucleus RNAseq.

Co-authors Ang Li, Jian Zeng, and Naomi Wray (University of Queensland and Oxford University) have conducted a comparison of a representative set of these methods (manuscript in preparation). Broadly, the methods are based on SNP-level regression (e.g., LDSC), gene-level regression (e.g., MAGMA-set), and cell scoring methods (e.g., scDRS). Methods that use SNP-level or gene-level regression methods differ in their specificity metrics to determine gene sets per cell type from the RNAseq date before integration with GWAS summary statistics. In contrast, scDRS takes a set of associated genes for a trait from the GWAS summary statistics into analyzes of the single-cell RNA-seq data. Li et al.[35] compared the performance of 9 representative methods: the approach used here (TDEP/S-LDSC)[17,18], MAGMA-set + EP, scDRS (using MAGMA-gene to select the top 1000 genes), sc-linker, and 5 CELLEX statistics (DET, GES, EP$_w$, NSI, ES$_\mu$). They evaluated these methods with respect to empirical data using the default settings from each method: 18 GWAS trait/cell type pairs for which no association was expected (e.g., proerythroblast cell type and asthma) and 18 GWAS trait/cell type pairs for which there was independent evidence for association (e.g., proerythroblast cell type and red blood cell count). Application of these methods to empirical data sets allowed estimation of false positive control and power in real-world scenarios.

In brief, Li et al.[35] found: (a) the method we use in this report (TDEP/S-LDSC) had power and false positive rates as good as or better than other methods; (b) MAGMA-set had somewhat higher power but at the cost of high false positive rates (indeed, we observed that MAGMA-set can yield markedly discrepant evidence for GWAS-cell type linkages – e.g., $P$-values 5–10 logs smaller than TDEP/S-LDSC); and (c) use of scDRS was constrained by computational burden – use of all 3.3 million cells for one GWAS trait took ~40 compute hours and 600 gb memory on a high-performance Linux cluster so that applying scDRS to the 36 primary GWAS was infeasible without down-sampling to a subset of nuclei. In addition, other authors have noted that TDEP performs well with respect to other expression metrics (Appendix 2, Fig. 3 in reference [26]).

**S-LDSC gene boundary expansion (±100 kb).** Expanding gene boundaries by ± 100 kb is often done and is generally consistent with the locations of promoters, enhancers, and eQTLs that impact gene expression[26,71]. We compared gene boundaries of ± 100 kb and ± 50 kb. For supercluster-level cell types and across the primary GWAS traits (Supplementary Data 1), the correlation between $\log_{10}$(enrichment-P) for ± 100 kb vs ± 50 kb was 0.988. We also calculated Cohen's kappa for the significance of the results (FDR correction) between the two choices of windows [using R::irr::kappa2()]. Even considering the conservative impact of significance thresholding, Cohen's kappa between the two windows was as high as 0.93. Given the small differences for these two gene expansion windows and to remain consistent with our prior papers[17,18], we used gene boundaries ± 100 kb.

**Conditional analysis of gene specificity overlap.** An important conceptual and practical issue is the degree of overlap in gene specificity between different cell types. We began by evaluating the overlap for all pairs of supercluster-level cell types. For the 435 unique supercluster pairs, the overlap was low (Jaccard index, JI, median 0.049, IQR 0.022-0.099, range 0.01−0.467). The lowest overlaps were for Amygdala excitatory-Vascular (JI = 0.01), Choroid plexus-Deep layer intratelencephalic (JI = 0.01), and Choroid plexus-Splatter (JI = 0.01). The greatest overlaps were for Deep layer intratelencephalic-Upper layer intratelencephalic (JI = 0.467), Eccentric medium spiny neuron-Medium spiny neuron (JI = 0.372), and Astrocyte-Bergmann glia(JI = 0.370). As shown in Supplementary Fig. 6 there were a few instances with a modest degree of clustering.

Although the overlaps were generally not marked over all pairs, all supercluster classes had at least one other class with potentially important overlap in specific genes; the maximum JI for each supercluster class had a median of 0.26 (IQR 0.19–0.33, range 0.12–0.47).

Therefore, the S-LDSC SNP-heritability enrichment of one supercluster class could be dependent on another class with overlapping TDEP genes. To examine such dependency, we performed conditional analyzes in a pairwise fashion. For each supercluster class, we added the TDEP genes of another class into the S-LDSC model and evaluated how that influenced the significance of enrichment for the supercluster of interest. If the result remained significant, it means that the enrichment for the supercluster of interest is statistically independent of the effect of the supercluster being conditioned upon. For instance, the signal for the "Upper layer intratelencephalic" class became non-significant after conditioning on the "Deep layer intratelencephalic" class (Fig. 3A), suggesting that the signal from the former was statistically dependent on the latter potentially due to overlap between TDEP genes. On the other hand, the signal of "CGE interneuron" supercluster was statistically independent from those of "MGE interneuron" and "LAMP5-LHX6 and Chandelier" despite overlap in TDEP genes (Supplementary Fig. 6).

**Conditional analysis of trait genetic overlap.** Another important perspective is the overlap in genetic risk between different traits. We started by evaluating the genetic correlations and the overlap between GWAS significant loci. More directly, we performed the GWAS-by-Subtraction analysis of genomicSEM[80] where we evaluated the common genetic effects that are specific to schizophrenia, conditioned on bipolar disorder, MDD, IQ, neuroticism, and educational attainment, respectively. We then applied S-LDSC SNP-heritability enrichment analysis at the supercluster level (Supplementary Fig. 7) for all five schizophrenia-specific GWAS. We further applied the same analysis at cluster levels for two schizophrenia-specific GWAS, conditioned on bipolar disorder (Fig. 3D) and IQ separately (Fig. 3E) which showcased the cell type signals shared between the two traits due to common (schizophrenia and bipolar disorder) and distinct (schizophrenia and IQ) mechanistic contributions.

**Gene set analysis.** Gene set analyzes were conducted using hypergeometric tests versus a background of 18,090 genes. Summarizing text from the beginning of the *Methods*: the 18,090 genes are based on the GENCODE primary assembly (GRCh38.p13, v35, 3/2020, hg38)[27,69]. In these analyzes, we focused on 18,090 genes that were protein-coding, mapped to canonical autosomes (chr1 to chr22), not in the extended MHC region (chr6:25-34 mb), and expressed in ≥1 of the 461 cell clusters. Hypergeometric *P*-values were FDR-corrected. The gene set analysis is performed for the TDEP genes of the 31 superclusters, as well as the top 25 clusters for schizophrenia. The major Gene Ontology (GO) terms in the gene sets enriched for the top 25 clusters for schizophrenia were summarized in treemaps using R::rrvgo. In addition to the Cellular Component (GO-CC) category (Fig. 3C), Supplementary Fig. 8 present the Biological Processes (GO-BP) and Molecular Function (GO-MF) domains, respectively.

**Density-based visualization of high-dimensional data.** To improve understanding of these high dimensional data, we applied UMAP to visualize these data in two dimensions and HDBSCAN to identify groupings within the UMAP projection[81,82].

**Analysis of brain anatomic dissections**
The dissection "HTHso" (supraoptic region of hypothalamus) had a high proportion of neocortical neurons, and "A35r" had a high proportion of non-neuronal cells[27], suggesting potential contamination or technical issues with the dissection depth or margins. We therefore excluded HTHso and A35r and focused on 104 anatomic dissections.

Anatomic dissections may contain highly heterogeneous cell type compositions[27], and the function of the same genes can differ between cell types. Therefore, we deem that S-LDSC is most appropriate at cell type levels (i.e., superclusters and clusters) and evaluate enrichment at the level of dissections (or, detailed brain regions) by integrating the cluster-level enrichment. For visualizing cell type components in Figure 4A, we calculated the proportion of each cell cluster per brain dissection as the number of cells in the cluster in the dissection divided by the total number of cells in the dissection. This number was scaled into deciles for plotting. Supplementary Data 9 presents the scaled proportion and the range and mean of the actual proportion.

Of note, neuronal cell types are dominantly enriched for the SNP-heritability of psychiatric disorders, we therefore focused on the neuronal cells in calculating dissection-level enrichment. Specifically, for each anatomic dissection, the proportion of each neuronal cluster was calculated as the number of cells in the cluster in the dissection divided by the total number of neuronal cells in the dissection. Next, for each dissection, we weighted the cluster-level enrichment Z-score (from S-LDSC) of each neuronal cluster by its proportion within the dissection, and calculated the weighted sum as the Z-score of enrichment for the dissection. Finally, *P*-value for each dissection was calculated based on the weighted-summed Z-score.

$$Z_{dissection} = \sum_{k}(pct_k \cdot Z_k) \tag{3}$$

where $k$ is a cluster, and $pct_k$ is the percentage of cluster $k$ in the dissection.

**Visualizing anatomic dissection results in a 3D human brain model**
The Allen Brain Atlas has a powerful 3D human brain model[83]. As the snRNA-seq dissections were performed according to the 2D Allen Brain Atlas[84], we had the opportunity to transfer the labels to the 3D map, although the names of the dissections were not completely matched between the 3D and 2D Atlas. We mapped 106 snRNA-seq dissections to 84 regions in the 3D brain model via expert curation (Supplementary Data 12). The matched regions were used for visualizing dissection-level SNP-heritability results of schizophrenia (Figure 4C, E) and to obtain 3D coordinates in the fMRI analysis (Fig. 5). Each 3D label was assigned a value as the -log₁₀(P) of scz2022 SNP-heritability enrichment of the corresponding anatomic dissection in the snRNA-seq dataset. If a 3D label corresponded to multiple snRNA-seq anatomic dissections, the most significant value was taken. Not all the cerebral cortical regions in the 3D model were sampled in the snRNA-seq dataset. Given our observation that the enrichment of scz2022 heritability was generally distributed across the cerebral dissections (Figure 4B), we assigned the unsampled neocortical regions the mean Z-score of all the sampled neocortical regions. For unsampled regions not in the cerebral cortex, we did not make any specific assumption, and their significance of enrichment was treated as missing. ITK_SNAP (v3.8.0, www.itksnap.org) was used to visualize the 3D model[85].

**Functional magnetic resonance imaging (fMRI)**
**Data sets for independent replication.** To evaluate our findings in the context of intrinsic functional connectivity, we compared patients with schizophrenia to neurotypical controls in two independent data sets. In data set 1, we included resting-state fMRI scans of 46 schizophrenia cases and 46 age- and sex-matched controls from the UCLA Consortium for Neuropsychiatric Phenomics LA5c Study (https://www.openfmri.org/dataset/ds000030/)[51]. From the public data set, 50 cases with schizophrenia and 127 healthy controls (age: 21–50 years) were acquired. Functional scans were pre-processed, head motion-corrected, and normalized to MNI space (resolution of 3 mm³)[86]. Frame-wise displacement (measuring the head motion during scanning)

showed significantly more head motion for cases than controls. To minimize group differences, participants with framewise displacement > 2 mm were excluded. Next, we gradually excluded the healthy controls with fewer head motions until the difference between the groups was no longer significant. In data set 2, the same quality control procession was applied to 72 patients and 75 controls (age: 18–65 years) from the Center for Biomedical Research Excellence (COBRE, https://fcon_1000.projects.nitrc.org/indi/retro/cobre.html), resulting in 54 schizophrenia cases and 54 age- and sex-matched controls.

**Define regions of interest (ROIs).** To define an initial search space in the fMRI imaging data, we started with 45 brain regions that were mapped to the 3D brain model and showed significant schizophrenia SNP-heritability enrichment at FDR ≤ 0.01 (Supplementary Data 13). We specified spherical regions of interest (ROIs) centering on the MNI coordinates (Supplementary Data 11) with a radius of 4 mm. Regions that were too small to be identified from the fMRI data were removed, and overlapping ROIs were selected based on higher enrichment of schizophrenia SNP-heritability. Specifically, substantial overlap was found between the ROIs of the amygdala corticomedial nuclear group (CMN), namely the medial nucleus (Me), the amygdalohippocampal area (AHA), and the cortical amygdaloid nuclei (specifically the anterior part, CoA); we used the CoA (highest significance of schizophrenia SNP-heritability enrichment) to represent CMN. Likewise, the lateral nucleus (La) of the amygdala was prioritized over the nearby, overlapping structures basolateral nucleus (BL) and basomedial nucleus (BM), to represent the basolateral nuclear group (BLN). The ambiens gyrus (AG), frontal agranular insular cortex (FI), and claustrum (Cla) were too small for ROI identification and thus removed. This resulted in 38 unique brain ROIs; taking laterality into account, we considered 76 regions (38 in each hemisphere) in the following analysis. The resting-state fMRI time-series of all voxels within each ROI were then averaged to obtain the regional time-series. We used Pearson's correlation of the functional time-series of the two ROIs as the connection, which was used as input data for the analysis.

**Prediction model (classifier) and key estimates.** We used a deep neural network classifier[87], Braph2[88], to distinguish cases from controls in the two data sets respectively. We randomly split the samples in each data set into five portions with similar size and balanced length of time series. In data set 1, we had 10 cases and 10 controls in portion 1, and 9 cases and 9 controls per portion in portions 2–5; and in data set 2, we had 10 cases and 10 controls in portion 1, 10 cases and 11 controls in portion 2, 11 cases and 11 controls in portions 3-4, and 12 cases and 11 controls in portion 5. We ran five folds of parallel analyzes in each data set. Per fold, we took one portion as the testing set and the rest (non-overlapping portions) as the training set to train the Braph2 classifier to distinguish cases from controls. Per fold, we adopted a recursive feature elimination strategy[89] to reduce the number of ROIs in a stepwise manner; in each run we obtained a network/model from the classifier and a measure to evaluate the contribution of each ROI to the model (detailed in the next paragraph), and the least contributing ROI was removed from the next run. In each run, we evaluated the trained model using the independent testing set and acquired the area under curve (AUC, of the receiver operating characteristic curve, ROC) as the evaluation measure. Outside the recursive feature elimination process, we evaluated the Core and Default Mode Networks as reference [29]. ROIs included in these two networks are listed in Supplementary Data 11.

Importantly, each model returned a feature importance (FI) per connection. The FI of all connections per ROI were then aggregated as the evaluation of the contribution of the ROI. The least contributing ROI was removed from the next run to realize the recursive feature elimination. Specifically, we used the cross-entropy loss function to calculate the model error in the optimization of the neural network

classifier. In each model, we performed 1000 permutations to evaluate the FI of all connections[90], where a single connection value was randomly shuffled 1000 times to establish the 95% confidence interval of the model error. If the model error of the original loss model fell outside the 95% confidence interval, the connection was considered contributing to the model, and its FI was calculated as:

$$FI = mean\_e\_perm / e\_orig \tag{4}$$

where $mean\_e\_perm$ is the averaged model error across the permutations, and $e\_orig$ is the original loss without any permutation.

**Interpretations at region and connection levels.** We used the runs that an ROI was preserved in the recursive feature elimination to evaluate the importance at region level, because more important ROIs in differentiating cases and controls were removed later during the process. As expected, hippocampal and amygdalar ROIs tend to be preserved to later runs in both fMRI data sets (Supplementary Figs. 9C, D), and the mean preserved runs of these regions had significant correlation between the two data sets (Fig. 5B). At connection level, we derived the aggregated FI matrix to evaluate the connections. Specifically, we scaled the FI matrix per model by dividing it by the maximum FI in the model, weighted the scaled FI matrix by the AUC of the model, and summed the processed FI matrices across models in the two fMRI data sets respectively (Supplementary Figs. 9E, F). We ranked and compared the connections by their FI (Fig. 5C–F) and performed hypergeometric tests to evaluate if the amygdalar or hippocampal connections were enriched in the top $n$ connections (Supplementary Figs. 9G–J), where $n$ were the even numbers from 2 to 5776. In total we ran 2888 hypergeometric tests to evaluate the enrichment per data set. In each data set, the name of ROIs in the top 1% connections were marked in red in Supplementary Fig. 9E, F and these ROIs were plotted on the dorsal view of the brain in Supplementary Fig. 9K, L.

### Cell type and evolutionary constraint

As described in Sullivan et al.[1], we used the Zoonomia alignment of 241 placental mammals to create a gene constraint metric. In comparing multiple different constraint metrics, the simplest metric appeared to be the best (cdsFracCons, the number of constrained CDS bases divided by the total number of CDS bases). cdsFracCons does not have the limitations of alternative measures (e.g., pLI is close to a dichotomy and LOEUF has a strong residual correlation with CDS size)[91,92].

### Reporting summary

Further information on research design is available in the Nature Portfolio Reporting Summary linked to this article.

## Data availability

Source data are provided with this paper. The data generated in this study are provided in the Supplementary Information and Source Data files. Source data are provided with this paper.

## Code availability

Code available at: https://github.com/Hjerling-Leffler-Lab/TDEP-sLDSC

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

## Acknowledgements

JHL was supported by the Swedish Research Council (Vetenskapsrådet, award 2018-00799), Swedish Brain Foundation (Hjärnfonden, award FO2018-0272) and European Research Council (SCHIZTYPE, grant agreement 819540). PFS was supported by the Swedish Research Council (Vetenskapsrådet, award D0886501) and the US National Institute of Mental Health (RO1s MH124871, MH121545, and MH123724). YL was supported by the European Research Council (SUBTREAT, grant agreement 101042183) and the US National Institute of Mental Health

(RO1 MH123724). SY was supported by the StratNeuro postdoctoral grant. JNL was supported by the Knut and Alice Wallenberg Foundation (KAW 2018.0152). We would like to acknowledge the International Headache Genetics Consortium (IHGC), International Suicide Genetics Consortium, Veterans Administration Million Veteran Program (MVP), MVP Suicide Exemplar Workgroup, for sharing the GWAS summary statistics on migraine and suicide attempt and suicide death (complete collaborator list in Supplementary Information). The fMRI data handling was enabled by resources in project NAISS sens2023024 provided by the National Academic Infrastructure for Supercomputing in Sweden (NAISS) at UPPMAX, funded by the Swedish Research Council through grant agreement no. 2022-06725. The fMRI data computations were performed at NSC Tetralith in project NAISS 2023/22-602 provided by the National Academic Infrastructure for Supercomputing in Sweden (NAISS) and PReSTO, funded by the Swedish Research Council through grant agreement no. 2022-06725 (NAISS) and 2018-06479 (PReSTO).

## Author contributions

Conceptualization (S.Y., Y.L., P.F.S., J.H.L.). Data Curation (S.Y., A.H., F.D., Y.L., P.F.S., J.H.L.), Formal Analysis (S.Y., A.H., F.D., Y.W.C., K.N.), Funding Acquisition (S.Y., J.N.L., Y.L., P.F.S, J.H.L.), Investigation (S.Y., A.H., F.D., Y.L., P.F.S., J.H.L.), Methodology (S.Y., A.H., F.D., Y.W.C., A.L., J.Z., N.R.W.), Project Administration (S.Y., P.F.S., J.H.L.), Resources (G.V., P.F.S., J.H.L.), Software (S.Y., A.H., F.D., Y.W.C.), Supervision (Y.L., P.F.S., J.H.L.), Visualization (S.Y., A.H., F.D.), Writing - Original Draft Preparation (S.Y., P.F.S., J.H.L.). Writing - Review & Editing (all authors).

## Funding

## Competing interests

PFS is a scientific consultant and shareholder for Neumora Therapeutics. The remaining authors declare no competing interests.
