## [Transparent Peer Review file · Nature Communications]

Connecting genomic results for psychiatric disorders to human brain cell types and regions reveals convergence with functional connectivity

Corresponding Author: Dr Jens Hjerling-Leffler

Version 0:

Reviewer comments:

Reviewer #1

(Remarks to the Author)

The authors have comprehensively addressed my suggestions and I recommend publication.

(Remarks on code availability)

As before, the code is well organized and commented.

Reviewer #2

(Remarks to the Author)

The authors have done a good job in revising the manuscript.

However, they failed to address the question about the neurobiological mechanisms underlying the link between statistical correlations of BOLD signal (resting state fMRI) with single cell RNAseq data. There is a large literature on the mechanisms of the BOLD signal, while the correlation maps derived from resting state fMRI are difficult to understand in terms of molecular mechanisms. This is a clear limitation and it is recommended to reduce the emphasis of this aspect, as it is a major limitation, even after the "replication".

(Remarks on code availability)

Reviewer #3

(Remarks to the Author)

Thank you for your responses

I remain unconvinced by the fMRI analysis, which is underpowered based on studies that you cite in your response and I do not think your manuscript needs these less convincing findings.

I am happy that you and the editor take a final decision on whether they should remain - I think this is otherwise an interesting and high-quality paper.

(Remarks on code availability)

I reviewed the code on the first submission

Point-to-point reply to review comments

We thank the reviewers for their time and helpful feedback. Below is the point-to-point reply to the comments from each reviewer.

The review comments were pasted in blue color. Our reply is in black color, and the changes in the manuscript are marked in red color.

Reviewer #1 (Remarks to the Author):

The authors have comprehensively addressed my suggestions and I recommend publication.

Reviewer #1 (Remarks on code availability):

As before, the code is well organized and commented.

Reply:

Thank you very much for the recommendation.

Reviewer #2 (Remarks to the Author):

The authors have done a good job in revising the manuscript.

However, they failed to address the question about the neurobiological mechanisms underlying the link between statistical correlations of BOLD signal (resting state fMRI) with single cell RNAseq data. There is a large literature on the mechanisms of the BOLD signal, while the correlation maps derived from resting state fMRI are difficult to understand in terms of molecular mechanisms. This is a clear limitation and it is recommended to reduce the emphasis of this aspect, as it is a major limitation, even after the "replication".

Reply:

Thank you for the comment and for pointing us to the interesting literature on the mechanisms of the Blood Oxygen Level-Dependent (BOLD) fMRI signal and its link with transcriptomic data. As you pointed out, the link between the BOLD signals and snRNA-seq data is not fully understood, despite a large literature.

In our paper, we aimed to validate the genomically prioritized regions by presenting the converging observation from a data-driven analysis of fMRI data. We have not analyzed changes in the local BOLD signal of the different areas, or the mechanism by this is generated which is beyond the scope of our research aim. Instead the functional connectivity analysis we applied focuses on the coordination between areas. In the discussion we now clarify this and suggest that the mechanism might be related to changes in synapse function which is supported by schizophrenia genetic findings.

To address your concern, we have discussed this as a **limitation** on **page 15, lines 32-42**:

“Finally, important limitations need to be considered when interpreting the results from the fMRI analysis. The molecular mechanism behind the fMRI blood oxygen level-dependent (BOLD) signals is not fully understood, but recent technological advancements in measuring local neural activity, neurovascular responses, and spatial transcriptomics will likely provide deeper insights^{67,68}. Our interpretation of the converging findings of the importance of brain regions from genomic and fMRI analyses does not imply a mechanistic link between the transcriptomic changes in neurons themselves and changes in local BOLD signal, but rather emphasizes changes in functional connectivity (i.e. coordination) between the brain areas which perhaps suggest changes in synaptic function/targeting. This interpretation is supported by the finding that schizophrenia genetics implicates synaptic function¹⁴. Furthermore, the sample size of the two independent fMRI data sets was limited given the case-control setting, and thus our initial results will have to be confirmed in larger data sets as they emerge.”

We also think that your suggestion of reducing the emphasis of the fMRI part will mitigate the misunderstanding (about our goal of showing the convergence rather than claiming

mechanistic relationship). As a result, we have **removed parts of the text** in the Discussion to reduce the emphasis (**page 15, Discussion > Implications for schizophrenia > the second paragraph**).

And we have also reorganized the Abstract (**page 1, lines 29-30**) and the first paragraph of the Discussion (**page 14, lines 10-14**):

“Using fMRI as an orthogonal modality, we show that a data-driven model trained to classify schizophrenia cases from neurotypical controls prioritizes connections between subcortical structures, particularly the amygdala and hippocampus. These findings provide support for convergence between different data-driven approaches on the brain regions identified through our genomic analysis.”

Reviewer #3 (Remarks to the Author):

Thank you for your responses

I remain unconvinced by the fMRI analysis, which is underpowered based on studies that you cite in your response and I do not think your manuscript needs these less convincing findings.

I am happy that you and the editor take a final decision on whether they should remain - I think this is otherwise an interesting and high-quality paper.

Reviewer #3 (Remarks on code availability):

I reviewed the code on the first submission

Reply:

We are thankful for your comment, and we agree that the sample size is a limitation despite the replicated observations. Thus, we have added this limitation in the Discussion (**page 15, lines 40-41**):

“Furthermore, the sample size of the two independent fMRI data sets was limited given the case-control setting, and thus our initial results will have to be confirmed in larger data sets as they emerge.”

We also thank you for recognizing the quality of the genetic and transcriptomic analyses, where we highlighted potential important brain regions, especially non-neocortical regions, for schizophrenia.

We agree that fMRI Data from large samples would be ideal and highly important. However, *"for rarer clinical conditions, amassing large samples is impossible ... Thus, small-sample neuroimaging will always be critical for studying the human brain"* (quoting the paper Marek et al. 2022). Therefore, in our minds the best solution was to replicate the finding in an independent data set.

With this part of the results, our aim is to provide external support to the genomically prioritized brain regions for schizophrenia, using an orthogonal data-driven modality. By showing the convergence, we propose a plausible link between these fields, which opens new questions for future studies, as we rewrote in the conclusion (**page 16, lines 12-13**). We therefore think this is an important addition to genetic and transcriptomic analyses, and we decided to keep it in the paper.